# SA6D: Self-Adaptive Few-Shot 6D Pose Estimator for Novel and Occluded Objects

**Ning Gao**[1,2]   **Ngo Anh Vien**[1]   **Hanna Ziesche**[1]   **Gerhard Neumann**[2]
[1]Bosch Center for Artificial Intelligence   [2]Autonomous Learning Robots, KIT
{ning.gao, anhvien.ngo, hanna.ziesche}@de.bosch.com
gerhard.neumann@kit.edu

**Abstract:** 6D pose estimation is one of the critical aspects in robotic manipulation. Most existing approaches have difficulties in extending predictions to scenarios where novel object instances are introduced, especially with heavy occlusions. In this work, we propose a few-shot pose estimation (FSPE) approach called SA6D on novel objects, which uses a self-adaptive segmentation module to identify the novel target object and construct a point cloud model of the target object using only a small number of cluttered reference images. Unlike existing methods, SA6D does not require object-centric reference images or any additional object information, making it a more generalizable and scalable solution across categories. We evaluate SA6D on real-world tabletop object datasets and demonstrate that SA6D outperforms existing FSPE methods, particularly in cluttered scenes with occlusions, while requiring fewer reference images.

**Keywords:** Contrastive Learning, 6D Pose Estimation, Self-Adaptation

## 1 Introduction

Accurately estimating the 6D pose of novel objects is critical for robotic grasping, especially for the tabletop setup. Prior work has investigated instance-level 6D pose estimation [1, 2, 3, 4], where the objects are predefined. Although achieving satisfying performance, these methods are prone to overfit to specific objects and suffer from poor generalization. Recently, several approaches [5, 6, 7, 8, 9, 10, 11, 12] have been proposed for category-level 6D pose estimation instead of specific objects. However, conditioning on specific object categories limits the generalization to objects from novel categories with strong object variations. Meanwhile, some approaches [13, 14, 15, 16, 17, 18] investigate generalizable 6D pose estimation as a few-shot learning problem, i.e., predicting the 6D pose of novel and category-agnostic objects given a few labeled reference images with the known pose of the novel object to define the object canonical coordinates. Although achieving promising results, these methods so far only work well on non-occluded and object-centric images, i.e., without the interference of other objects. This limits the generalization to real-world scenarios with multiple objects in cluttered and occluded scenes. Furthermore, additional object information is required such as object diameter [14], mesh model [18, 13], object 2D bounding box [17] or ground-truth mask [16, 19], which is not always available for novel object categories. Our method aims to enable a fully generalizable few-shot 6D object pose estimation (FSPE) model.

In summary, we identify three primary challenges that are not adequately addressed by the current state-of-the-art methods [14, 16, 17, 18]: i) The category-agnostic 6D pose estimation in cluttered scenes with heavy occlusions is performing poorly. ii) The object-centric reference images from cluttered scenes are cropped by ground-truth segmentation or bounding box of the target object, which limits the generalization in real-world scenarios. iii) The requirement of extensive reference images covering all different view-points is not practical.

7th Conference on Robot Learning (CoRL 2023), Atlanta, USA.

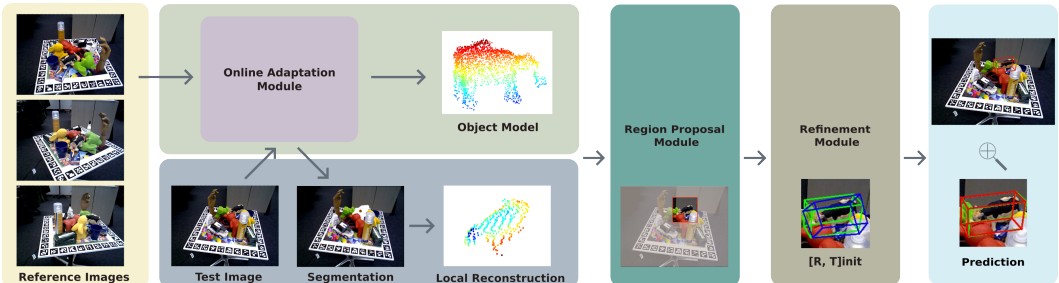

Figure 1: We present a generalizable and category-agnostic few-shot 6D object pose estimator using a small number of posed RGB-D images as reference. Compared to existing methods, our approach provides robust and accurate predictions on novel objects against occlusions without requiring re-training or any object information.

To address the aforementioned challenges, we propose a robust self-adaptive 6D pose estimation approach called SA6D. As shown in Fig. 1, SA6D uses RGB-D images as input since i) depth images are normally easy to obtain along with RGB images in robotic setup, and ii) depth images reveal additional geometric features and can improve the robustness of prediction against occlusion. SA6D employs an online self-adaptative segmentation module to contrastively learn a distinguishable representation of the novel target object from the reference images of cluttered scenes. Meanwhile, a canonical point cloud model of the object is constructed from the depth images. After the online adaptation, the segmentation module is capable to segment the target object from new images and construct the local point cloud from depth. Incorporating geometric features from the extracted point cloud, a region proposal module crops the test image by localizing the target object. With the cropped test image and the reference images, we employ Gen6D [14] to first predict an initial pose using visual input, followed by a refinement module using the induced geometric features.

Our work focuses on the scenario with tabletop objects used for robotic manipulation. Our primary contributions are summarized as follows:

- SA6D is fully generalizable to new datasets without requiring any object or category information such as ground-truth segmentation, mesh model, or object-centric image. Instead, only a limited number of RGB-D reference images with the ground-truth 6D pose of the predicted object are needed.

- A self-adaptive segmentation module is proposed to learn a distinguishable representation of novel objects during inference.

- SA6D significantly outperforms current state-of-the-art methods against occlusion in real-world scenarios while trained entirely on synthetic data.

## 2   Related Work

**Category-Level 6D Object Pose Estimation.** Methods for generalizable 6D object pose estimation can be divided into category-specific and category-agnostic models. For the category-specific estimation, Wang *et al.* [5] first propose a canonical representation for all possible object instances within a category using Normalized Object Coordinate Space (NOCS). However, inferring the object pose by predicting only the NOCS representation is non-trivial given large object variations [21]. To tackle this problem, Tian *et al.* [22] account for intra-categorical shape variations by explicitly modelling the deformation from shape prior to the object model, while CASS [6] generates 3D point clouds in the canonical space using a variational autoencoder (VAE) [23]. FS-Net [8] proposes a shape-based model using 3D graph convolutions and a decoupled rotation mechanism to further reduce the feature sensitivity to the color variations. Wang *et al.* [7] predict the relative 6D pose between two consecutive images using a category-based keypoint matching model. Chen *et al.* [9] employ a VAE-based generator to learn a categorical prior and update the prior with online rendering

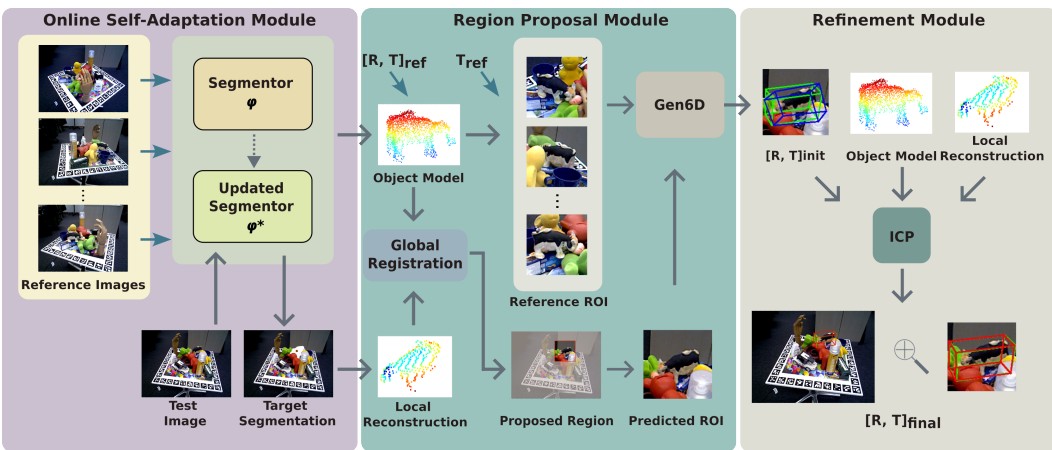

Figure 2: **Overview.** SA6D includes three modules: i) The *online self-adaptation module* discovers and segments the target object (*milk cow*) from a cluttered scene giving a few posed RGB-D images as reference. Subsequently, the canonical object point cloud model from the reference images and the local model from the test image are constructed based on the segments. ii) The *region proposal module* outputs a robust region of interest (ROI) of the target object against occlusion by incorporating visual and geometric features. A coarse 6D pose is then estimated by comparing the cropped test and reference images using Gen6D [14] and iii) further fine-tuned using ICP [20].

w.r.t. the test image. Recently, Fu *et al.* [11] facilitate the generalization by collecting a large-scale dataset with object-centric RGB-D videos called Wild6D. Based on Wild6D, Zhang *et al.* [12] propose to learn the dense 2D-3D correspondences between the 2D image pixels and the categorical shape prior while the final 6D pose is computed by the least-square-fitting algorithm [24]. Similar to our work, UDA-COPE [25] employs self-supervised training while TTA-COPE [26] addresses the source-to-target domain gap using test time adaptation. Nevertheless, these methods require a manually defined categorical prior for training and therefore are limited to generalize across categories. In contrast, our method learns 6D pose estimation in a category-agnostic manner.

**Category-Agnostic 6D Object Pose Estimation.** Category-agnostic pose estimation can be formulated as a few-shot learning problem [15]: During inference, the model can generalize and predict the pose of novel objects given a few images with known poses as reference. LatentFusion [16] and iNeRF [19] employ the neural rendering technique [27] to refine the predicted pose based on a latent representation obtained from the reference images while a segmentation of the object is required as input. FS6D [17] extracts features from both the reference images and test images followed by a prototype matching algorithm to obtain the point-wise correspondences. OnePose [28] and OnePose++ [29] build an object model from a single RGB video and employ feature mapping between the test image and the object model, which are not end-to-end and deviate from the few-shot domain. Furthermore, all the aforementioned methods require object-centric images for either reference or test images. In contrast, Gen6D [14] is applicable for cluttered scenes where both reference and test images contain multiple objects, although it struggles with occlusion. Our work is inspired by Gen6D and exploits the geometric information of the target object to enable robust prediction against occlusion.

**Unseen Object Segmentation.** Recently, several approaches have been proposed to close the gap between learning unseen object segmentation from synthetic datasets and real world datasets [30, 31, 32, 33]. Xie *et al.* [34] propose to learn a two-stage segmentation model by separately leveraging RGB and depth information in a hierarchical way, where the model is fully trained on the synthetic data. UCN [35] proposes to learn from RGB and depth images jointly and generate pixel-wise feature embeddings. To enable a generalizable 6D pose estimation in cluttered scenes, we design a self-adaptive module to generate target-object-oriented segmentation model using UCN as a base segmentor.

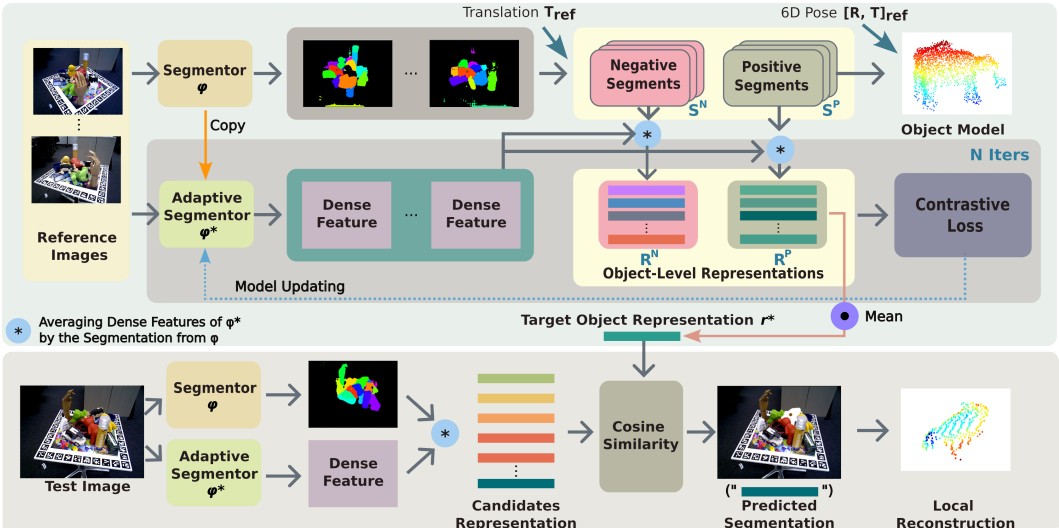

Figure 3: **Online self-adaptation module**. A pretrained segmentor $\varphi$ is first applied on reference images to predict segmentations. Meanwhile, an adaptive segmentor $\varphi*$ is initialized from $\varphi$. With the ground-truth translation of the target object in the reference images $\mathrm{T}_{\mathrm{ref}}$, the object center can be reprojected to the image. For each reference image, one segment is chosen as a positive sample if it includes the reprojected object center while the remaining segments are considered as negative samples. Subsequently, an object-level representation of each segment is computed by averaging the pixel-wise dense features from $\varphi*$. A contrastive loss is then applied over the positive and negative object representations and updates $\varphi*$ iteratively. After adaptation, $\varphi*$ generates the target object representation $r*$ by averaging over all positive representations from reference images. Given a test image, we obtain the representation of each candidate segment in the same way and compute the cosine similarity between each candidate and $r*$, where the most similar candidate is chosen as the segment of the target object. Meanwhile, the canonical and local object models are computed based on the segments and depth images.

## 3 Method

SA6D is comprised of three parts, i.e., an online self-adaptation module (OSM) for target object segmentation from cluttered scenes, a region proposal module (RPM) to infer the region of interest (ROI) for the target object against occlusion, and a refinement module (RFM) to refine the predicted 6D pose of the target object using both visual and the inferred geometric features. The proposed pipeline is shown in Fig. 2.

### 3.1 Online Self-Adaptation Module

To alleviate the dependence on prior object information and object-centric reference images, and improve the prediction against occlusions, it is essential to build a model which can discover the objects from the cluttered scene and identify the occluded target object from other objects. To achieve this, we design a self-adaptive segmentation module which is updated only during inference in a self-supervised manner given posed reference images, where no retraining is needed.

In particular, we employ the segmentation model from Xiang *et al.* [35] as our base segmentor $\varphi$, which segments all instances of each image by clustering the pixel-wise features using the mean-shift algorithm [36]. Examples of predicted segmentation are shown in Fig. 3. Given the ground-truth translation $\mathrm{T}_{\mathrm{ref}}^{i} \in \mathbb{R}^3$ of the target object in the $i$-th reference image, we can reproject the object center on the image plane and select the segment, which includes the reprojected object center, as a positive target segment $s_i^P$, while the remaining segments are considered as negative segments $S_i^N = \{s_1^N, ..., s_K^N\}$. $K$ denotes the number of negative segments in each reference image. Given

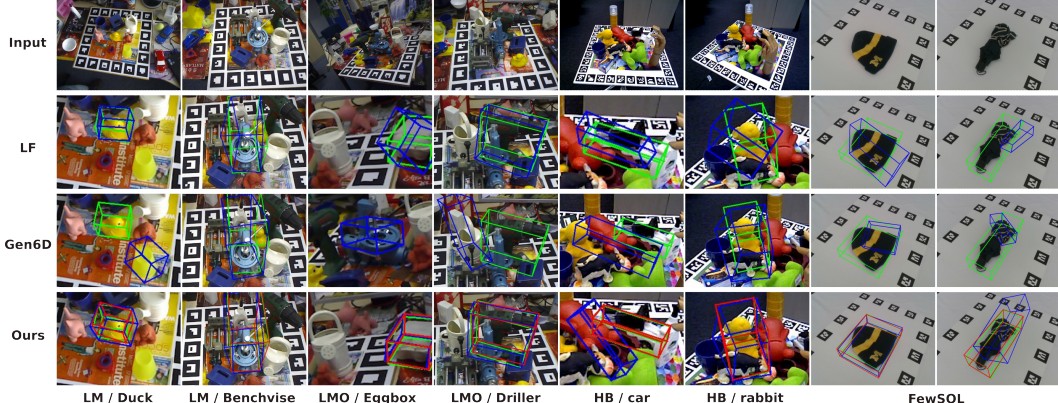

Figure 4: **Qualitative results.** The green bounding box denotes the ground-truth pose and blue denotes the prediction. In SA6D, blue denotes prediction before refinement while red is the final prediction.

$M$ reference images, we obtain a positive set of target object segments $S^P = \{s_1^P, ..., s_M^P\}$ and a negative set of segments $S^N = \cup_{i=1}^M S_i^N$.

The adaptive segmentor $\varphi^*$ is initialized by copying the parameters of $\varphi$ and used to generate distinguishable representations between the target object and the remaining objects, not for generating the segmentation. The positive and negative object-level representations, $R^P$ and $R^N$ are computed by averaging the pixel-wise dense features of $\varphi^*$ while grouping by each segment from $S^P$ and $S^N$. Based on the positive and negative representation sets $R^P$ and $R^N$, $\varphi^*$ is updated iteratively using a contrastive loss [37]. Specifically, for each positive pair $r_i^P, r_j^P \in R^P$, the loss is computed as

$$l_{ij} = -\log \frac{\exp(\mathrm{sim}(r_i^P, r_j^P)/\tau)}{\sum_{r' \in R^N \cup \{r_j^P\}} \exp(\mathrm{sim}(r_i^P, r')/\tau)}, \tag{1}$$

where $\tau$ is a hyper-parameter and set to $0.07$, sim denotes the cosine similarity between two representations. The loss is summed over all combinations of the positive pairs from $R^P$ and backpropagated through $\varphi^*$. After adaptation, $\varphi^*$ generates the target object representation $r*$ by averaging over all positive representations $r* = \mathrm{mean}(r_1^P, ..., r_M^P)$. Note that $R^P$ and $R^N$ are updated along with $\varphi^*$ simultaneously.

Given a test image, the candidate segments are obtained in the same way from $\varphi$ followed by computing the representations from $\varphi*$. By comparing the cosine similarity between the candidates and the target object representation $r^*$, the candidate with the highest similarity score is selected as the segment of the target novel object.

**Object Model Reconstruction.** We reconstruct the object model from the reference images by computing the partial point clouds for each positive segment and transfer them to the canonical coordinates given the ground-truth 6D pose $[R, T]_{\mathrm{ref}}$. The combination of partial point clouds obtained from the reference images assembles a coarse geometric model of the object. For inference, we obtain a partial point cloud model (local reconstruction) using the predicted target segment from the test image.

### 3.2  Region Proposal Module

The region proposal module combines 2D image features and the geometric features from induced point cloud model. The idea is to improve the robustness against clutter and occlusion in comparison to Gen6D [14]. The region of interest (ROI) denotes a squared area where the target object is located. While Gen6D can predict the ROI of novel objects, it suffers from occlusion since the prediction depends purely on the visual similarity between the reference and test images.

| Method | GT-Mask | Ref. Num | LineMOD [39] | | | | | | | LineMOD-OCC [40] | | | | | | | HomeBrewedDB [41] | | | | |
|---|---|---|---|---|---|---|---|---|---|---|---|---|---|---|---|---|---|---|---|---|---|
| | | | eggbox | duck | benchvise | cam | cat | glue | avg. | driller | eggbox | duck | glue | ape | can | avg. | cow | flange | car | rabbit | avg. |
| Gen6D [14] | ✗ | 20 | 0.63 | 0.30 | 0.45 | 0.29 | 0.25 | 0.26 | 0.36 | 0.09 | 0.02 | 0.07 | 0.03 | 0.12 | 0.21 | 0.09 | 0.36 | 0.15 | 0.15 | 0.52 | 0.30 |
| SA6D (ICP only) | ✗ | 20 | 0.53 | 0.31 | 0.37 | 0.25 | 0.21 | 0.17 | 0.31 | 0.17 | 0.16 | 0.10 | 0.08 | 0.14 | 0.22 | 0.14 | 0.23 | 0.17 | 0.20 | 0.44 | 0.26 |
| SA6D (wo/ RPM) | ✗ | 20 | 0.63 | 0.47 | 0.50 | 0.37 | 0.36 | 0.38 | 0.45 | 0.19 | 0.15 | 0.13 | 0.10 | 0.17 | 0.28 | 0.17 | 0.37 | 0.19 | 0.21 | 0.61 | 0.35 |
| SA6D (wo/ RFM) | ✗ | 20 | 0.57 | 0.36 | 0.45 | 0.34 | 0.29 | 0.26 | 0.38 | 0.15 | 0.08 | 0.09 | 0.04 | 0.10 | 0.28 | 0.12 | 0.39 | 0.12 | 0.20 | 0.55 | 0.32 |
| SA6D | ✗ | 20 | **0.73** | **0.73** | **0.55** | **0.50** | **0.47** | **0.72** | **0.62** | **0.45** | **0.26** | **0.30** | **0.21** | **0.32** | **0.53** | **0.35** | **0.62** | **0.35** | **0.33** | **0.78** | **0.52** |
| Gen6D | ✗ | 64 | 0.74 | 0.40 | **0.73** | 0.65 | 0.65 | 0.53 | 0.62 | 0.27 | 0.09 | 0.23 | 0.03 | 0.11 | 0.50 | 0.21 | 0.38 | 0.06 | 0.49 | **0.78** | 0.43 |
| SA6D | ✗ | 64 | **0.80** | **0.84** | **0.73** | **0.80** | **0.84** | **0.75** | **0.79** | **0.44** | **0.41** | **0.38** | **0.31** | **0.33** | **0.66** | **0.42** | **0.72** | **0.49** | **0.72** | 0.69 | **0.66** |
| LF [16] | ✓ | 20 | 0.61 | **0.61** | 0.68 | 0.65 | **0.72** | **0.78** | 0.67 | 0.28 | 0.01 | 0.00 | 0.18 | **0.45** | 0.17 | 0.18 | 0.33 | 0.00 | 0.00 | 0.16 | 0.12 |
| SA6D (wo/ RFM) | ✓ | 20 | 0.56 | 0.32 | 0.54 | 0.30 | 0.26 | 0.29 | 0.38 | 0.10 | 0.06 | 0.08 | 0.04 | 0.24 | 0.14 | 0.11 | 0.41 | 0.13 | 0.15 | 0.54 | 0.31 |
| SA6D | ✓ | 20 | **0.68** | 0.58 | **0.80** | **0.73** | 0.72 | **0.78** | **0.72** | **0.33** | **0.26** | **0.29** | **0.30** | 0.19 | **0.45** | **0.30** | **0.58** | **0.17** | **0.44** | **0.76** | **0.49** |

Table 1: Evaluation of ADD-0.1d on LineMOD, LineMOD-OCC, and HomeBrewedDB datasets against category-agnostic baselines.

| Method | ADD-0.1d | ADD-0.3d | ADDs-0.1d | ADDs-0.3d |
|---|---|---|---|---|
| LF [16] | 0.1162 | 0.1738 | 0.1299 | 0.1907 |
| Gen6D [14] | 0.3571 | 0.6399 | 0.6399 | 0.7530 |
| SA6D (wo/ RFM) | 0.4018 | 0.7292 | 0.6964 | **0.8780** |
| SA6D | **0.5595** | **0.7887** | **0.8393** | **0.8780** |

Table 2: Evaluation on FewSOL [42] dataset over 336 objects using 8 reference images.

| Method | $IOU_{0.5}$ | 5°2cm | 5°5cm | 10°5cm |
|---|---|---|---|---|
| CASS [6] | 0.01 | 0.0 | 0.0 | 0.0 |
| Shape-Prior [22] | 0.33 | 0.03 | 0.04 | 0.14 |
| DualPoseNet [43] | 0.70 | 0.18 | 0.23 | 0.37 |
| RePoNet [11] | **0.71** | 0.30 | 0.34 | **0.43** |
| SA6D | 0.65 | **0.37** | **0.40** | 0.42 |

Table 3: Evaluation on Wild6D [11] dataset against category-level baselines.

Instead of requiring the object diameter in Gen6D, we estimate the object diameter $\hat{d}$ from the reconstructed object model. With the reconstructed object point cloud model from inference images and the partial point cloud model from the test image, we estimate an initial translation $T_{init}$ using global registration, i.e., using RANSAC to first match the points between the two models followed by the fast point registration proposed by Zhou *et al.* [38]. With $\hat{d}$ and the estimated depth $T_{init}^z$, the ROI scale is computed by $s = \hat{d} * f/T_{init}^z$ where $f$ is the camera focal length. Meanwhile, the ROI position $[u, v]$ is calculated by $u = T_{init}^x * f/T_{init}^z$ and $v = T_{init}^y * f/T_{init}^z$. As shown in Fig. 2, we can accurately predict the ROI against occlusion using the geometric information without interference from the environment. Similar to the test image, the cropped reference images are obtained given the reconstructed object model and ground-truth pose $[R, T]_{\text{ref}}$. Subsequently, we employ the pretrained Gen6D detector to predict an initial pose based on the visual input.

### 3.3 Refinement Module

Based on the reconstructed geometric features of the object, we employ Iterative Closest Point (ICP) [20] and use the output of Gen6D as an initial pose, which helps alleviate the problem of local optimal in ICP. In our experiments, we found this is particularly useful for rotation estimation where the global point registration struggles.

## 4 Experiments

We employ two baselines that are most relevant to our work on category-agnostic unseen objects, namely LatentFusion (LF) [16] and Gen6D [14]. Besides the input image, LatentFusion requires ground-truth segmentation of the target object while Gen6D requires the object diameter as input. In contrast, our method does not require any additional information. We also compare SA6D against category-level SOTA methods which use RGB-D as input. It is good to note that SA6D is not trained on any specific category while all category-level baselines are trained and tested on the objects within the same category.

### 4.1 Datasets and Metrics

**Evaluation datasets.** We use four datasets for evaluation against category-agnostic methods, namely LineMOD [39], LineMOD-OCC [40], HomebrewedDB [41] and FewSOL [42]. None of these datasets is used during the training phase. LineMOD (LM) includes annotations of 15 test objects in cluttered scenes without occlusion while LineMOD-OCC (LMO) and HomebrewedDB

(HB) have heavy occlusion. FewSOL includes 336 real-world objects and 9 RGB-D images for each object from different viewpoints, where we randomly select 8 images as references and test on the remaining image. FewSOL includes large-scale object variations but without occlusion or clutter. Furthermore, we compare against category-level methods on Wild6D [11] which is a RGB-D video dataset including 5 object categories.

**Training datasets.** The base segmentor is trained fully on the synthetic Tabletop Object Dataset (TOD) generated by Xie *et al.* [44] and the pretrained Gen6D model uses the rendered images from ∼2000 ShapeNet [45] models and 1023 Google Scanned Object by Wang *et al.* [46]. Note that only synthetic datasets are used for training.

**Evaluation metrics** We use the average distance (ADD) [39] to evaluate the object points after being transformed by the ground-truth and predicted pose. ADD-0.1d (ADD-0.3d) denotes the accuracy of the predictions with an average distance below $10\%$ ($30\%$) of the object diameter. ADD-S is used in FewSOL dataset due to the large amount of symmetric objects, where the average distance is computed based on the closest point. For comparison against category-level methods, we employ the same BOP [47] metrics as used in RePoNet [11]. $n°$, $m\ cm$ denotes the accuracy when both rotation error is less than $n°$ and translation error is less than $m\ cm$.

## 4.2 Results and Discussion

**Comparison against category-agnostic methods.** As shown in Tab. 1, although baselines show promising results on LineMOD dataset, they perform poorly and cannot generalize on occluded datasets (LineMOD-OCC and HomeBrewedDB). In contrast, without requiring ground-truth segmentation or object diameter, SA6D significantly increases the performance over all datasets, especially under the circumstances where fewer reference images are given or the objects are occluded. Furthermore, without ground-truth segmentation, SA6D still outperforms LatentFusion on occluded datasets by a large margin. Tab. 2 shows SA6D is able to generalize on large object variations while LatentFusion cannot generalize even without occlusion. We find that LatentFusion requires high-quality depth images and more reference images to reconstruct the latent representation, and works poorly on flat objects. Examples are shown in Fig. 4. Furthermore, SA6D demonstrates superior performance against Gen6D even without using the geometric features in the refinement module (RFM). The reason is, Gen6D struggles with localizing the target object in FewSOL dataset since the evaluated objects in FewSOL dataset are close to the camera and occupy a larger area than the dataset used for training, indicating a poor generalization of Gen6D on out-of-distribution data. In contrast, the region proposal module (RPM) used in SA6D alleviates the problem.

**Ablation study on components of SA6D.** To evaluate the importance of different components in SA6D, we conduct an ablation study by removing the region proposal module (*wo/ RPM*), the refinement module (*wo/ RFM*), and remove both by only using the global point cloud registration between the reconstructed global and local object model (*ICP only*). The results are shown in Tab. 1. The performance decrease of *ICP only* indicates that classic point cloud registration between partial and global point clouds is often stuck at a suboptimal position. The performance drop on *wo/ RFM* demonstrates the importance of the induced geometric features and the notable performance drop on LineMOD-OCC and HomeBrewedDB without using RPM shows its crucial role against occlusion. In Fig. 5c, we show an example of a test image and visualize the pixel-wise visual similarity between reference and test images on top, where higher brightness indicates higher visual similarity. RPM is capable to localize the target object (cow) while Gen6D depends purely on visual similarity and selects a wrong object.

**Accuracy vs Reference Number.** We report the ADD-0.1d w.r.t. the number of reference in Fig. 5a on HomebrewedDB/cow. Increasing the number of reference images overall benefits all methods except LatentFusion, which sometimes shows degradation in performance because a heavily occluded reference image can drastically alter the implicit representation in the latent space due to the employed online rendering. Notably, SA6D performs consistently better than baselines and shows reasonable prediction with a one-shot reference image.

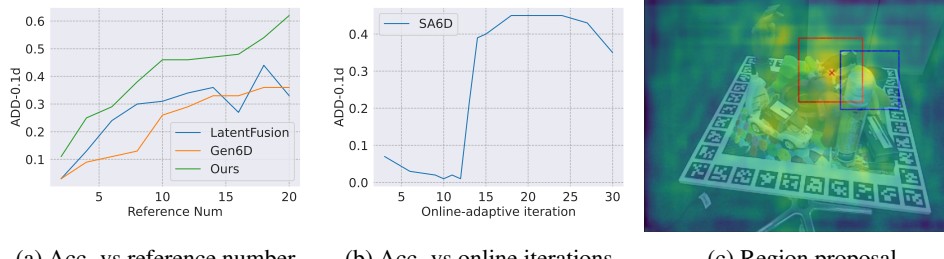

| (a) Acc. vs reference number | (b) Acc. vs online iterations | (c) Region proposal |

Figure 5: Analysis of the number of (a) reference and (b) online iterations. (c) An example of proposed ROI from SA6D (red) and Gen6D (blue), the red cross denotes the target object.

**Analysis of Online Self-Adaptation.** The performance of SA6D w.r.t. the number of iterations in the online self-adaptation module is shown in Fig. 5b on LineMOD-OCC/driller. At the beginning, SA6D performs poorly since the segmentor $\varphi*$ cannot learn and differentiate the representation of the target object from others, which also leads to a performance decrease. After 12 iterations, with a learned distinguishable target object representation, SA6D gains significant improvement. With more iterations, the performance decreases again since the updated segmentor $\varphi*$ starts overfitting to the reference images. We prevent overfitting by automatically stopping updating $\varphi*$ with a defined threshold to the contrastive loss in Eq. (1). In our experiments, we set the threshold to 0.01 over all datasets.

**Comparison against category-level methods.** Tab. 3 demonstrates the comparison against category-level SOTA methods on Wild6D dataset. Although SA6D is not trained specifically on each category, it overall achieves competitive performance and even outperforms baselines using the strict criteria (*5°2cm*), which indicates SA6D can predict more accurate poses than all baselines. In the appendix, we also visualize the predictions of SA6D and RePoNet [11] for comparison.

**Discussions.** We find that our online self-adaptation module is robust against false positive samples and is able to learn a correct target-oriented representation and shows remarkable performance against severe occlusion and truncation. Moreover, SA6D provides explainable confidence scores by computing the cosine similarity among the candidate segments. We also tried replacing ICP with a learning-based method, namely RPM-Net [48]. However, the prediction is always stuck at the sub-optimum. Nevertheless, we believe SA6D can be further improved with future development in the area of segmentation and learning-based point could registration, which is not the main focus of this work. The inference running time on a single image costs ∼0.93s in total on Nvidia Tesla V100 for SA6D. More discussion on dynamic scenes are shown in the appendix A.3.

### 4.3 Limitations

Our work does not consider deformable or articulated objects, especially for cases where reference and test images have drastic shape diversity. Another notorious concern is predicting transparent objects where sensors are often failed to capture depth information. Recent work on depth completion for transparent such as Zhu *et al.* [49] can alleviate the problem. Furthermore, our method requires an accurate and generalizable base segmentor $\varphi$. Although SA6D achieves promising results in most cases for tabletop objects, the under- and over-segmentation behaviors still limit the performance. Moreover, a more generalizable learning-based registration method between partial and global point clouds would be an interesting direction to replace ICP.

## 5 Conclusion

We propose an approach that can efficiently and robustly predict the 6D pose of novel objects with heavy occlusions while not requiring any object information or object-centric images. We hope our approach can facilitate generalizable 6D object pose estimation in robotic applications.

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

In appendix, we show additional qualitative results on the predicted target object segmentation including severe occlusion and truncation, qualitative results on the final 6D pose estimation against Gen6D, comparison between ICP and learning-based point cloud registration method, additional ablation studies, and further explanation and analysis on existing methods compared to our method, e.g., the selection of reference images, the effort of annotation, and the practical use case. We also submit a video introducing SA6D in the supplementary material.

# A    Additional Results

## A.1    Gen6D without ground-truth object diameter.

In Tab. 4, we demonstrate that using the object diameter as input is a strong prior knowledge which limits the generalization of Gen6D, by fixing the diameter over all objects with two different values, namely $10\ cm$ and $50\ cm$. Without ground-truth diameter, Gen6D cannot generalize well on any of the datasets.

| Diam. (m) | Dataset | | | |
| --- | --- | --- | --- | --- |
| | LM | LMO | FewSOL | HB |
| 0.1 | 0.06 | 0.06 | 0.04 | 0.10 |
| 0.5 | 0.16 | 0.05 | 0.00 | 0.19 |
| GT | **0.35** | **0.08** | **0.36** | **0.30** |

Table 4: Evaluation on Gen6D with different object diameters as prior knowledge. Results are averaged over objects for each dataset.

## A.2    Compare ICP with learning-based point cloud registration algorithm

We show a few predicted examples of a state-of-the-art learning-based point cloud registration model, namely RPM-Net, on the LineMOD-OCC/driller in Fig. 6. RPM-Net is prone to the local optimal position for 6D pose estimation, especially for rotation. In our experiments, ICP is more robust to unseen objects.

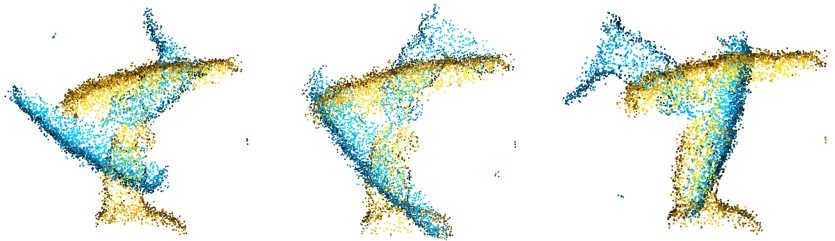

Figure 6: Examples of using RPM-Net for point cloud registration instead of ICP. The yellow point cloud denotes the reconstructed object point cloud model and the blue one denotes the prediction after transformation using the predicted pose from RPM-Net. Better overlapping between two point clouds indicates better performance. RPM-Net cannot generalize on unseen objects and is prone to get stuck in local optima.

## A.3    Results on dynamic scenes.

We have conducted the evaluation on dynamic scenes where we sample 20 images from LineMOD-OCC (LMO) as reference images and evaluate query images from LineMOD (LM) which includes different background and objects. Thus, the reference and query images are from different scenes with changing light conditions and configuration of surrounding objects. The results are shown in Tab. 5. SA6D achieves competitive results compared to the original setup (first line in the table).

Note that this is still not a fair comparison for our method since we use reference images with occluded target objects, which makes it more difficult to reconstruct the object model. However, this experiment demonstrates the capability of our method on query images from novel scenes.

| Ref. image | Query image | eggbox | duck | cat | glue | avg. |
|------------|-------------|--------|------|-----|------|------|
| LM | LM | 0.73 | 0.73 | 0.47 | 0.72 | 0.66 |
| LMO | LM | 0.70 | 0.71 | 0.46 | 0.72 | 0.64 |

Table 5: Evaluation on LineMOD using LineMOD-OCC as reference.

### A.4 SA6D is robust to false positive samples in reference

Using reprojected object center to select positive segments sometimes leads to a false positive sample given the target object center is occluded. An example is shown in Fig. 7a, in which a wrong segment (*yellow rabbit*) is selected as a positive sample for the target object (*milk cow*). However, we find that our online self-adaptation module is robust against false positive samples and is able to learn a correct target-oriented representation. Moreover, SA6D provides explainable confidence scores by computing the cosine similarity between each segment representation and the target object representation. Fig. 7b shows an example of the predicted target (*milk cow*) segments with reasonable induced confidence score though wrong positive samples are given in the reference set.

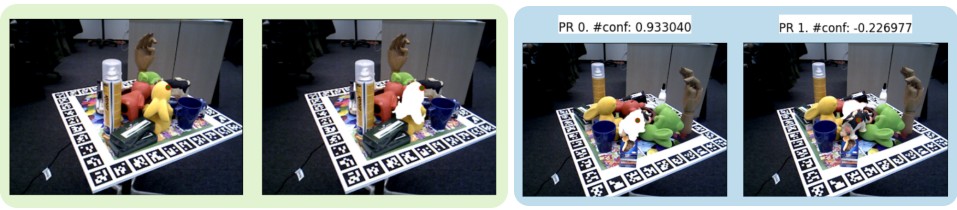

(a) False positive sample.  (b) Robust prediction.

Figure 7: Discussion. (a) A false positive sample is selected given the reprojected center of the target object (*milk cow*) is occluded by another object (*yellow rabbit*). Nevertheless, (b) SA6D provides robust prediction with explainable confidence scores.

### A.5 SA6D demonstrates remarkable performance against severe occlusion and truncation

We show superior performance of SA6D on challenging scenes with severe occlusion and truncation in Fig. 8, where the input images, predicted segmentations from the base segmentor $\varphi$, ground-truth segmentation of target object based on the reprojected object center, and three predicted candidates with the highest predicted confidence scores are given on each column from left to right. The selected segments are marked in white color. The confidence score *conf* denotes the cosine similarity between the candidate segment representation and the target object representation $r^*$. The *conf_seg* is computed by dividing the confidence scores between the first and second most similar segment candidates w.r.t. the target object representation. Thus, it can be used in crucial scenarios if the prediction is uncertain among different segments. Note that in Fig. 8a, our method is able to differentiate the target object segment while the provided ground-truth segmentation points to a wrong segment due to the center of target object is occluded.

### A.6 Robust and explainable confidence score of the online self-adaptation module

We show more results on the predicted segmentation of the online self-adaptation module in Fig. 9 on LineMOD dataset, Fig. 10 on LineMOD-OCC, and Fig. 11 on HomebrewedDB. Some candidates in Fig. 11 with white background indicate the background segments are selected.

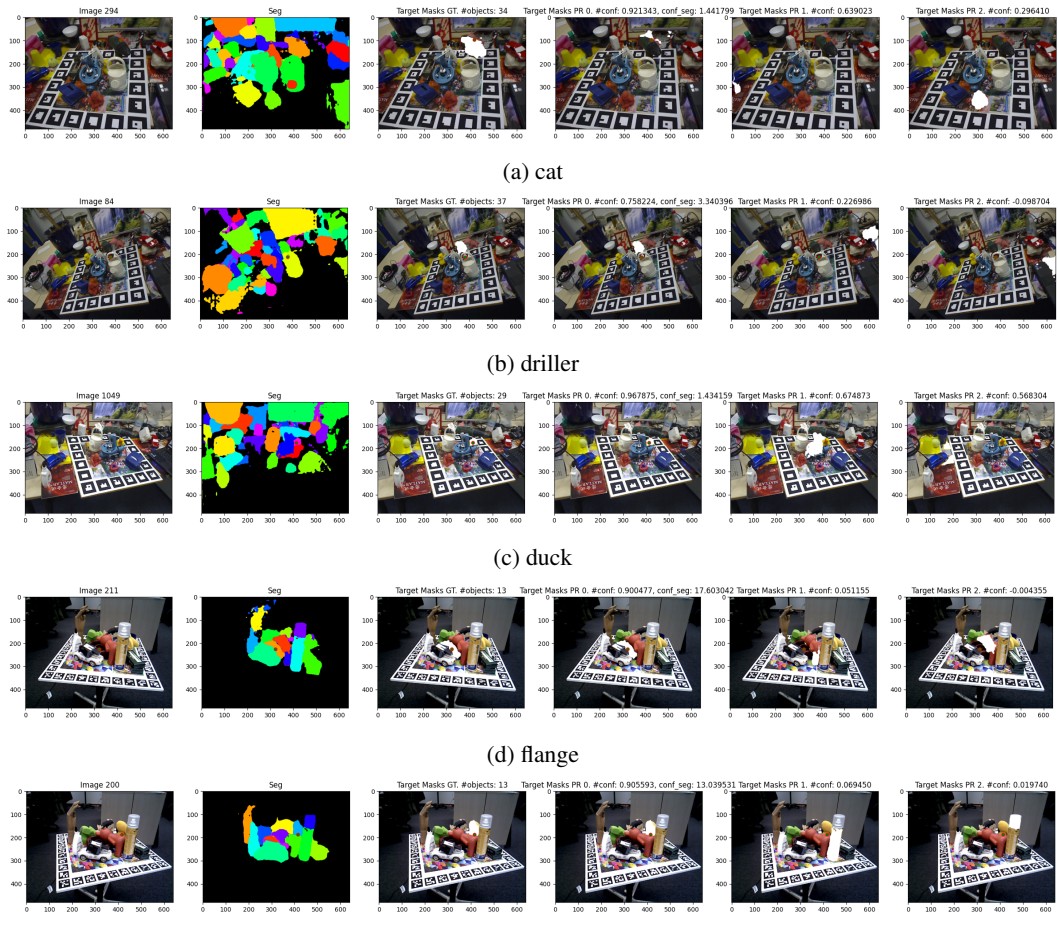

(a) cat

(b) driller

(c) duck

(d) flange

(e) yellow rabbit

Figure 8: Online-Adaptation results on challenging scenes against severe occlusion and truncation. Three candidates with the highest confidence scores are visualized in order.

### A.7  More Qualitative Results

We show more qualitative results of the 6D pose prediction and compare our method with Gen6D on LineMOD dataset in Fig. 13, LineMOD-OCC in Fig. 14, HomebrewedDB in Fig. 15 and FewSOL in Fig. 16. The comparison on Wild6D dataset between SA6D and category-level SOTA method RePoNet is shown in Fig. 17.

### A.8  Failure Cases

We show the examples in Fig. 12 where using ICP leads to a worse prediction than without using ICP in the refinement module. Results are evaluated on the FewSOL dataset, indicating future work on generalizable and learnable point cloud registration is essential to further improve the performance.

## B  Additional Explanation

### B.1  Selection of Reference Images

Regarding the selection of the reference images on the LM, LM-O and HB datasets, the original Gen6D selects 64 reference images from a predefined set of images with farthest point sampling (FPS) to make sure that the view distributes evenly among the reference images. We follow the

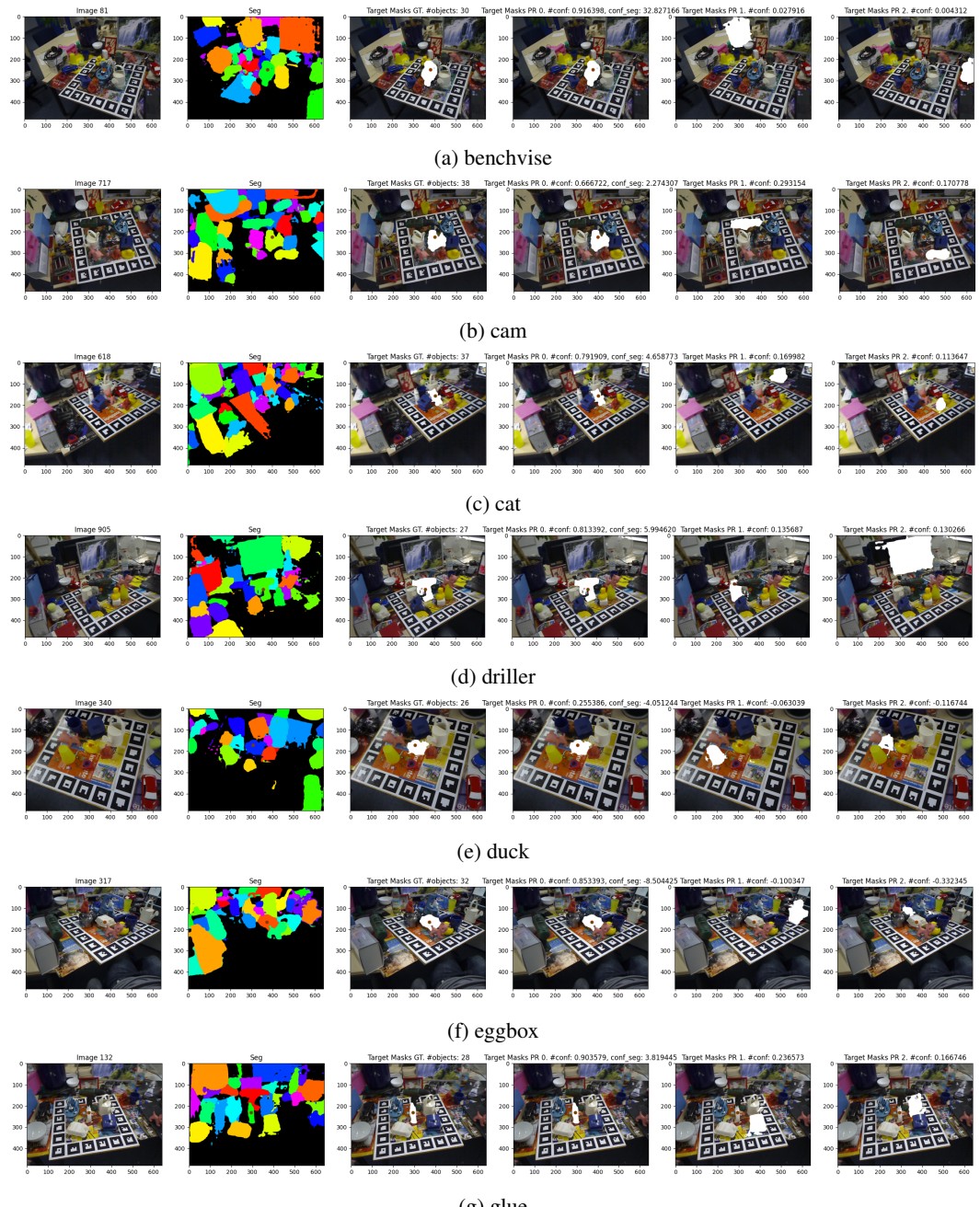

Figure 9: Robust prediction of target segmentation on LineMOD. Three candidates with the highest scores are visualized in order.

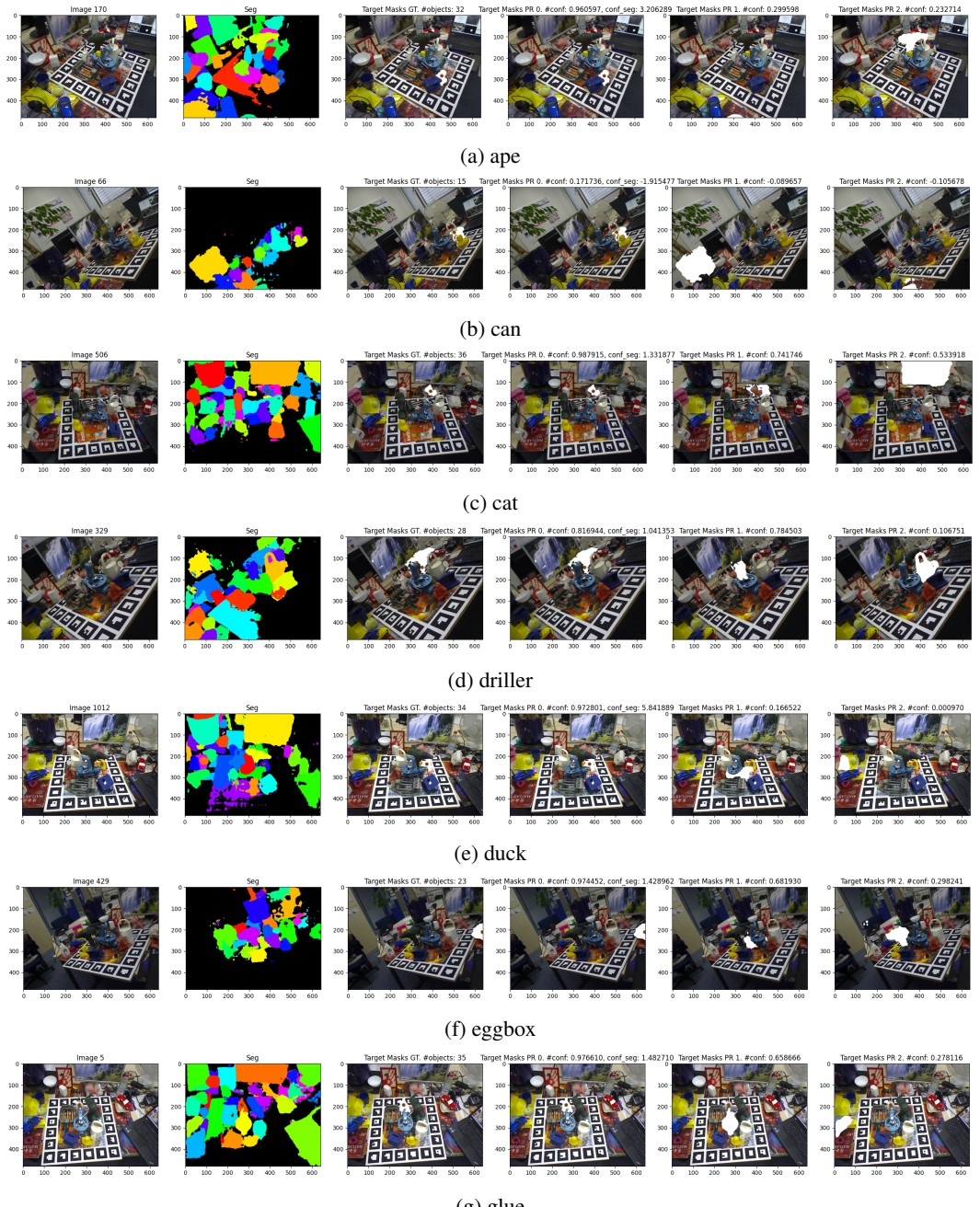

Figure 10: Robust prediction of target segmentation on LineMOD-OCC. Three candidates with the highest scores are visualized in order.

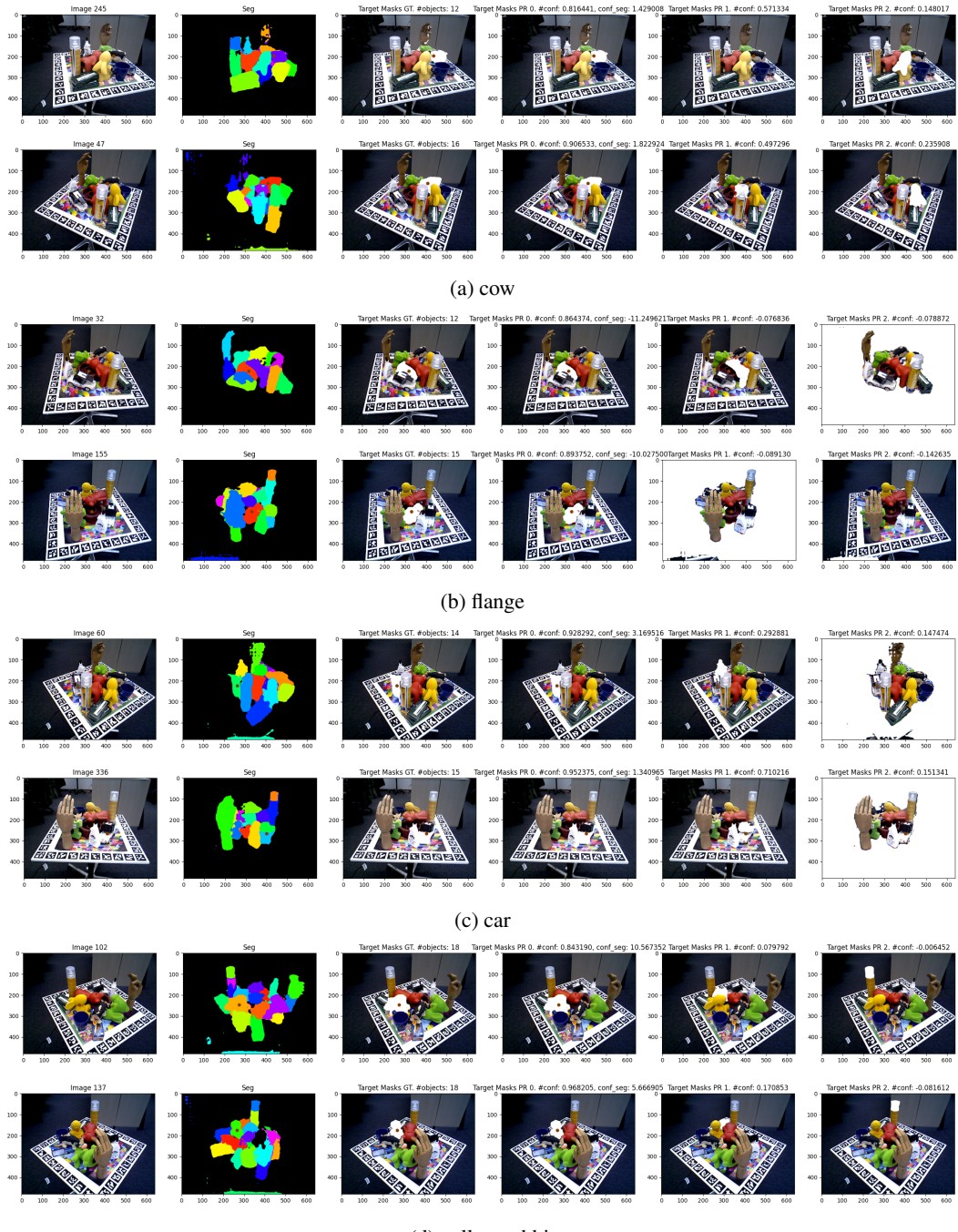

Figure 11: Robust prediction of target segmentation on HomebrewedDB. Three candidates with the highest scores are visualized in order.

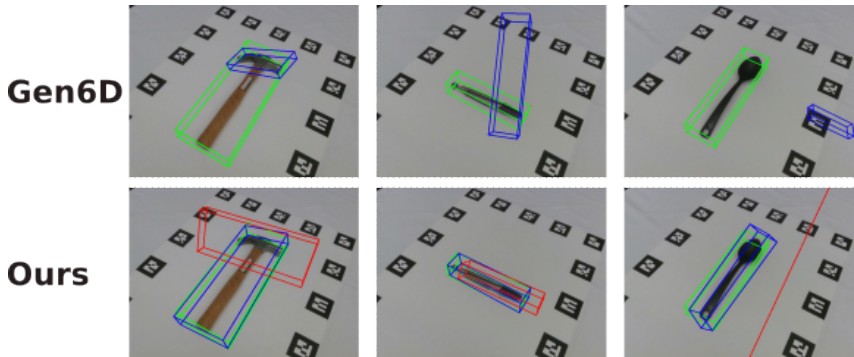

Figure 12: Failure cases. Using ICP in the refinement module leads to a worse prediction than the initial prediction. The green bounding box is the ground-truth pose. The blue bounding box denotes the prediction in Gen6D and the prediction before using ICP in the refinement module in our method while the red one denotes the prediction after ICP.

same setup when all models are evaluated with 64 reference images. However, it is not efficient to sample 64 images and it is often the case that the reference images are not distributed evenly in the real-world. Therefore, we also evaluate all methods by randomly selecting 20 reference images from the dataset, which significantly increases the task difficulty but is more realistic and plausible because it is not always obtainable to collect reference images that could cover all viewpoints.

## B.2 Comparison with FS6D and Model-Based Models

Similar to LatentFusion, FS6D [17] also requires object-centric reference images with ground-truth segmentations for cluttered scenes. Considering that its code is not published and we could not reproduce its results, we hence excluded FS6D in our comparisons. Meanwhile, we cannot add the model-based methods [4, 3, 2, 1] into comparison due to their limitation, i.e., the model-based methods can only be applied on the specifically trained object and cannot work in our setup where the results are evaluated on new objects. Also, it is unfair to compare them with our work if we train the model-based methods on the new objects. Moreover, the FewSOL dataset contains only 9 images for each object, which is insufficient to train the model-based methods. Considering all these limitations of the model-based methods, it is also one of our motivations to work on this paper.

## B.3 Effort of Annotation Compared with Prior Work

The annotation of a limited number of reference images requires human effort. However, the effort of annotation is also essential in prior work [8, 4, 3, 2, 22, 1, 5] where thousands of annotated images are required for every single object or category. Category-agnostic methods such as our method tremendously reduce human effort by requiring only a small number of annotations. Still, similar to Gen6D and LatentFusion, it is necessary to have a small number of posed reference images for an unseen object to set the canonical object coordinates to further determine the object rotation w.r.t. the camera. Importantly, our method does not require any additional effort compared to existing methods.

## B.4 Practical Use Case

Our method can be used in the lifelong robot item picking/sorting in industry. Each time when a new product comes in, the robot only needs to sample a small number of images with ground-truth 6D pose between the new product and the camera by moving the robot arm around the new product where the camera is mounted on the robot arm and the other objects together with the new product are placed on a calibrated picking plate. The pose between the camera and the new product is easily obtainable since the pose of the camera and new product w.r.t. the robot base coordinates

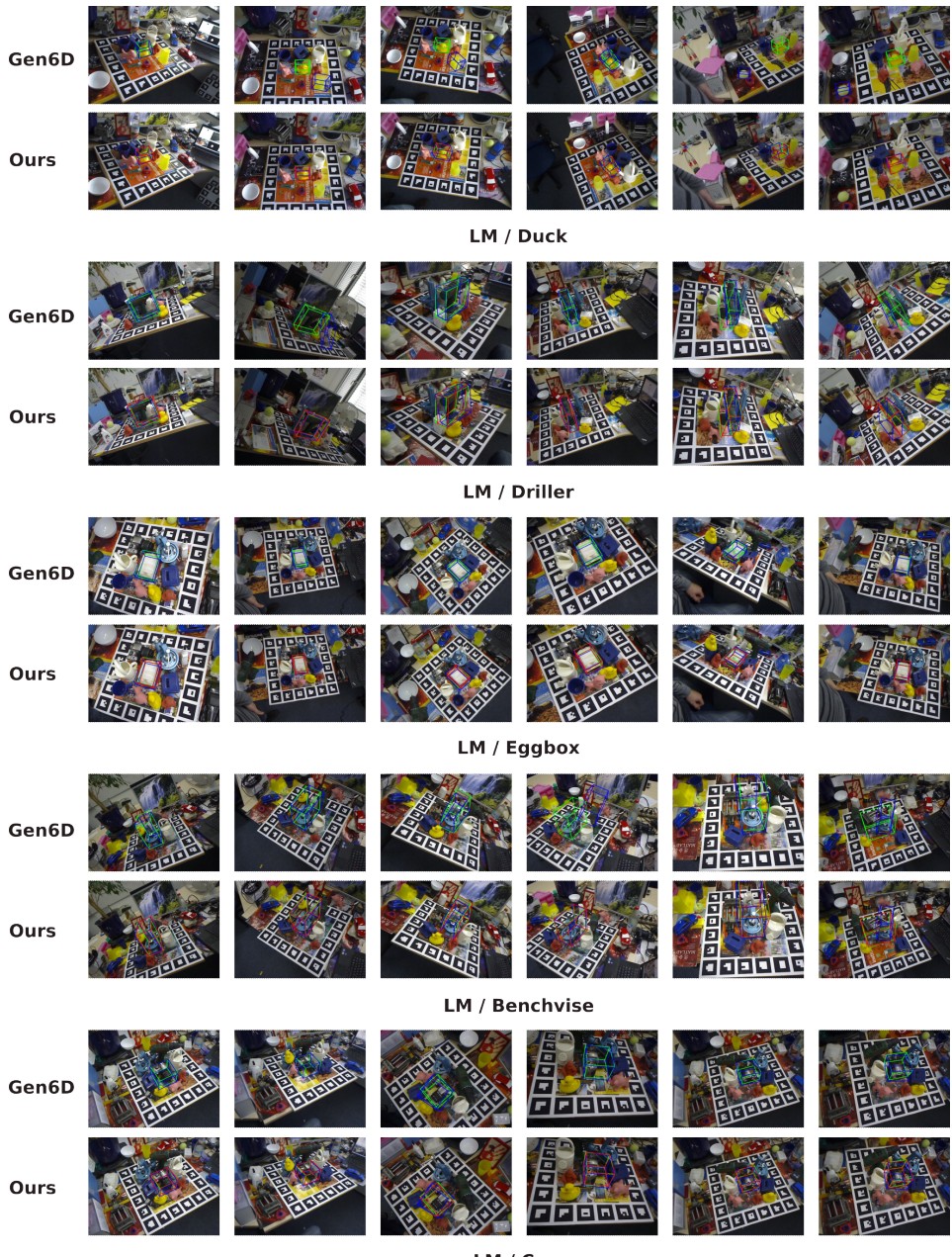

Figure 13: Prediction on LineMOD dataset with 20 reference images. The green bounding box is the ground-truth pose. The blue bounding box denotes the prediction in Gen6D and the prediction before using ICP in the refinement module in our method while the red one denotes the prediction after ICP.

are known. Thus, the whole system can be fully automatic and does not require further training for new products.

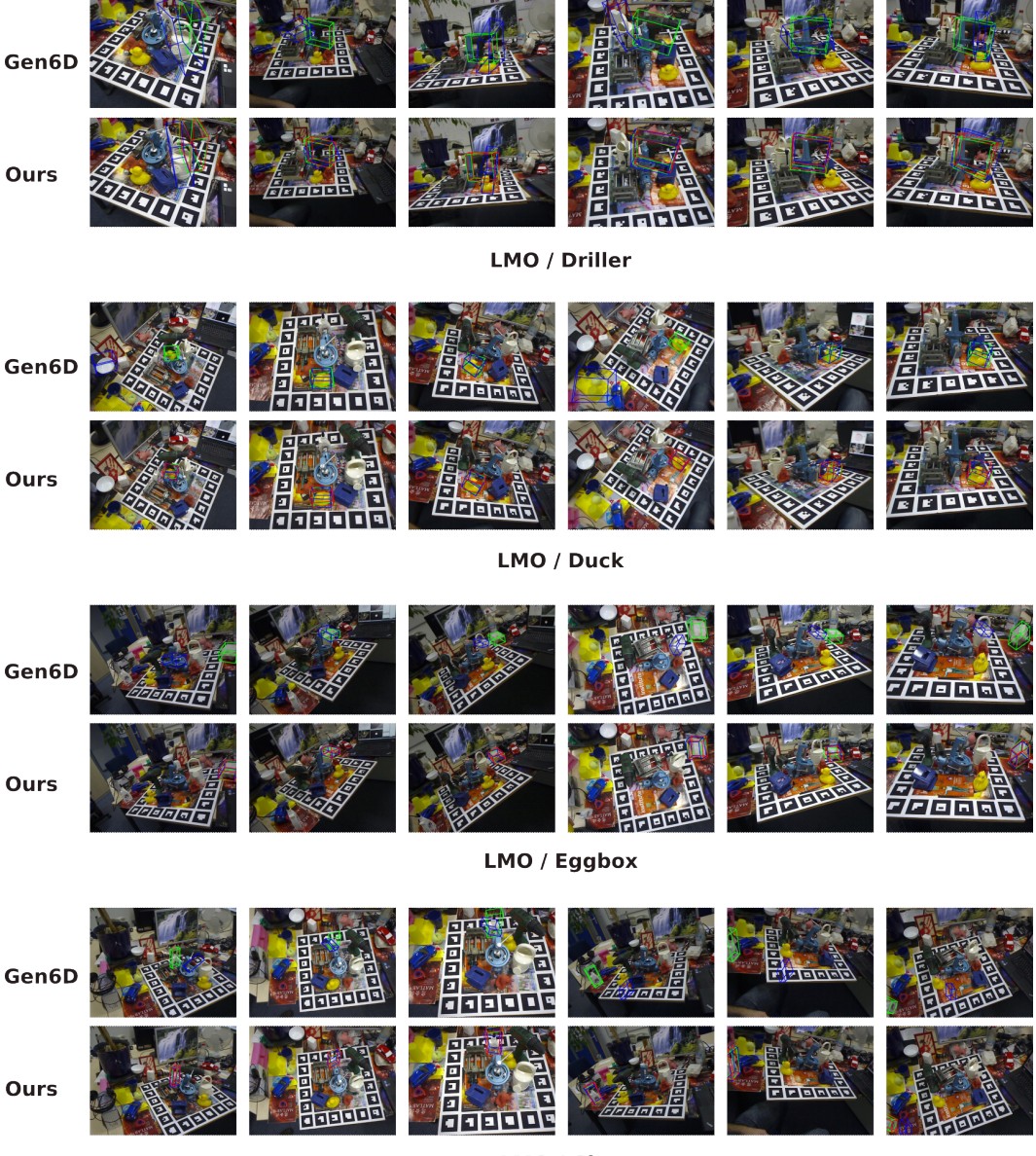

Figure 14: Prediction on LineMOD-OCC dataset with 20 reference images. The green bounding box is the ground-truth pose. The blue bounding box denotes the prediction in Gen6D and the prediction before using ICP in the refinement module in our method while the red one denotes the prediction after ICP.

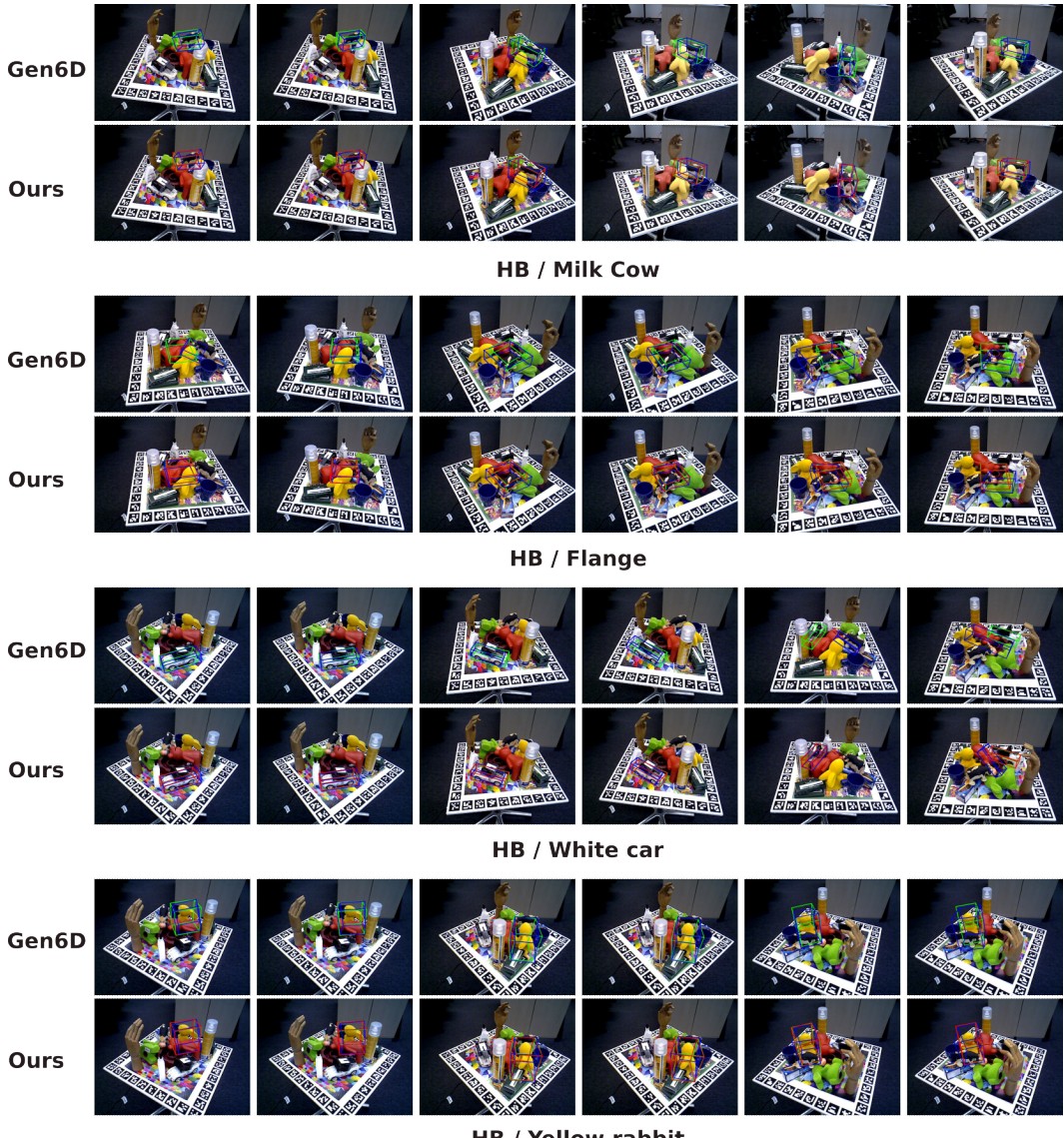

Figure 15: Prediction on HomebrewedDB dataset with 20 reference images. The green bounding box is the ground-truth pose. The blue bounding box denotes the prediction in Gen6D and the prediction before using ICP in the refinement module in our method while the red one denotes the prediction after ICP.

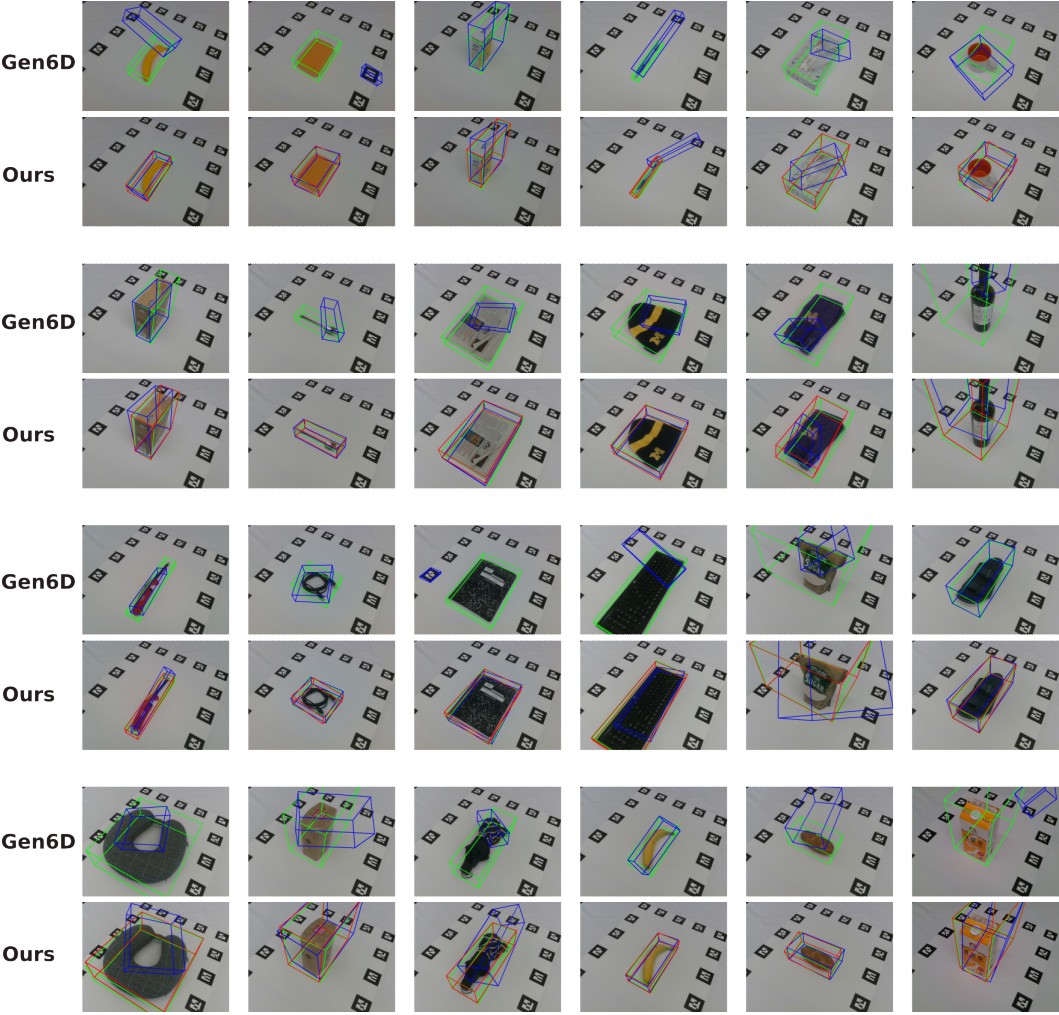

Figure 16: Prediction on FewSOL dataset with 20 reference images. The green bounding box is the ground-truth pose. The blue bounding box denotes the prediction in Gen6D and the prediction before using ICP in the refinement module in our method while the red one denotes the prediction after ICP.

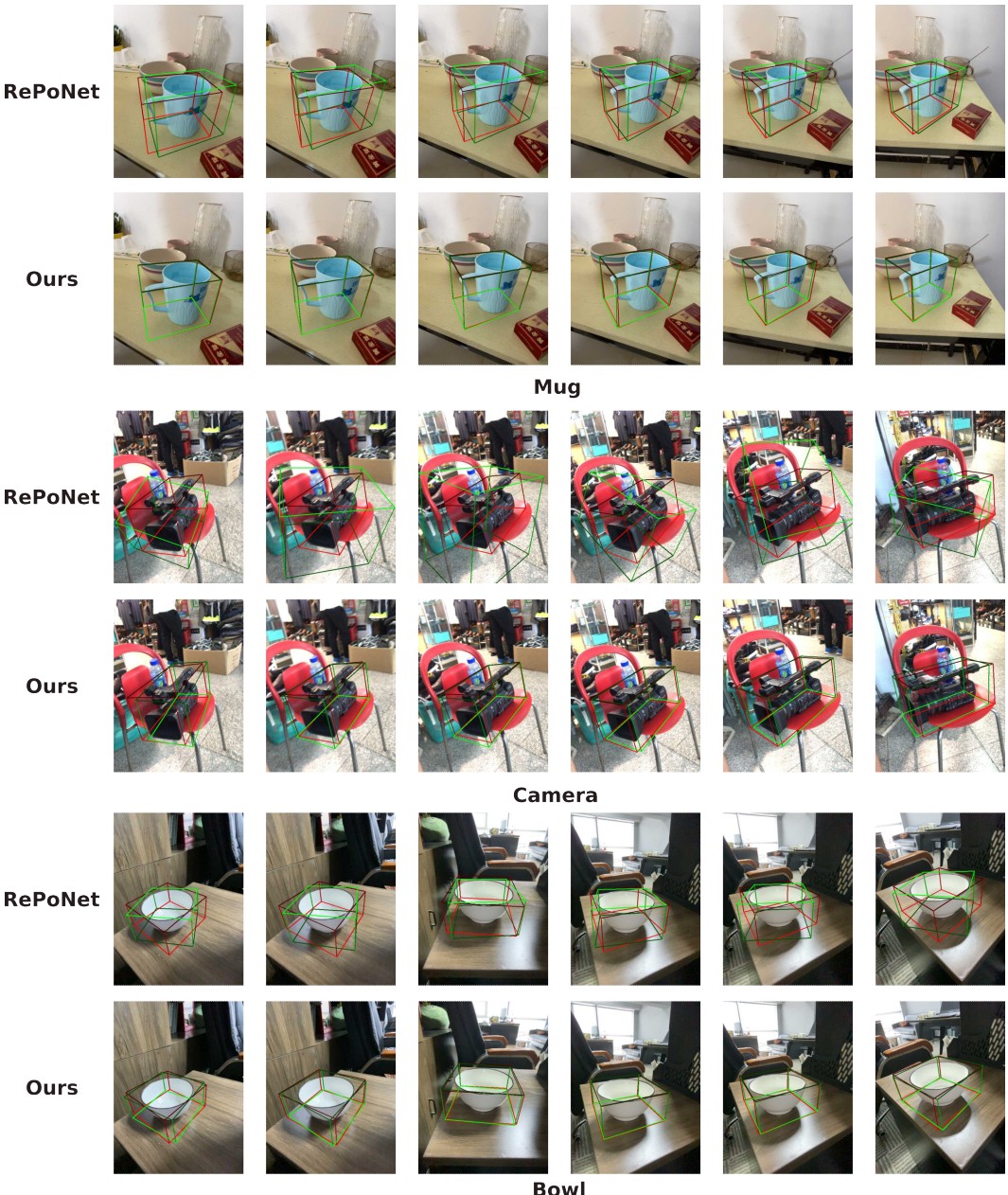

Figure 17: Prediction on Wild6D dataset with 20 reference images. The red bounding box is the ground-truth pose and the green bounding box denotes the prediction.

