# OpenReview forum: "SA6D: Self-Adaptive Few-Shot 6D Pose Estimator for Novel and Occluded Objects"
_robot-learning.org/CoRL/2023/Conference — CoRL 2023 Poster_

### Official Review · Reviewer_L1f1 · 2023-07-14

**Confidence:** 4
**Originality:** Good
**Technical Quality:** Very Good
**Clarity Of Presentation:** Very Good
**Impact:** 3

**Recommendation:**

Strong Accept: I recommend accepting the paper and will argue for my recommendation even if other reviewers hold a different opinion.

**Review:**

## Summary
### Strengths
- Improved performance and robustness to occlusion and clutter w.r.t. state-of-the-art methods.
- Thorough experiments and ablations.
- Very good quality of the manuscript.
- The supplementary material (particularly the video) is useful and facilitates the understanding of the main manuscript.

### Weaknesses
- The method is not extremely novel, in the sense that it combines pre-existing tools (although this combination is not trivial and results in improved performance).
- Compared to previous few-shot approaches, the method assumes the availability of depth information.

## Detailed review
### Quality and clarity
- The problem that the paper addresses is clearly stated and motivated, and an extensive list of related work is provided.
- The experiments are thorough, with evaluations on several standard datasets and with multiple types of baselines (both few-shot methods and RGB-D category-level methods) and ablations on the model components. All my main questions regarding the influence of the difference components were answered through these experiments.
- The quality of the manuscript, of the visualization and the supporting material is in general high, with a very limited amount of typos and mostly clear and linear language (cf. "Suggestions and minor questions" below).


### Originality and significance
- Although the method does not present groundbreaking new ideas, it combines pre-existing tools in a effective way, resulting in a clear improvement over previous methods.
- No experiments from a real-world setup are presented, but the method is evaluated in several real-world benchmarks.
- I find particularly interesting the simple, but smart idea, behind the selection of the mask for the object of interest in the reference images, by using the known extrinsics. I also appreciate the discussion in Sec. A.3 about how such a strategy might fail, but the confidence score can still compensate for it.

### Suggestions and minor questions
**Language**

- The abstract is probably the main part for which I would suggest rephrasing. In particular, the sentence "[...] meaningful robotic manipulation [...] one of the critical aspects" sounds vague and a bit clunky (e.g., what does "meaningful" mean exactly? "Critical" for what?). Similarly, the term "continuously [introduced]" (L4) can have a fairly specific (and diverse) meaning in the literature, so when reading it I would expect the paper to put emphasis for instance on the "open-set" problem, or potentially on some continual learning problems, which is not done in the rest of the paper.
- Minor: Throughout the paper, "i.e." could be replaced with the macro "\ie".
- L124: "copying $\phi$" -> "copying **the parameters of** $\phi$".
- L128: $\phi*$ -> $\phi^*$ or $\phi^\star$.
- Caption of Table 1: "HomwBrewedDB" -> "HomeBrewedDB".
- In all its occurrences (header of Table 1, L223, caption of Fig. 5, etc.), I would recommended replacing "reference number"/"reference" with "number of reference images".
- L203: Fix "cannot _even though without_ [...]". Also, on the same line, the tense of "found" is not coherent with the rest of the paragraph.
- L225: "[...] except that LatentFusion sometimes [...]" -> "[...] except LatentFusion, which sometimes [...]".
- L246: "replace" -> "tried replacing" or "experimented replacing" (tense coherency).
- L250: "costs" -> "is".
- Fig. 7 in the Suppl.: "Even though" -> "Nevertheless".
- L58 Suppl.: Missing "the case" after "It is often".
- L65 Suppl.: "exclude" -> "excluded", "We" -> "we"

**Notation and formatting**
- L123: Technically, $S^N$ should be defined as $\bigcup_{i=1}^M S^N_i$, since the $S^N_i$s are sets. A space is missing after the word "segments".
- L155-157: It could be clarified that $T^x_{init}$, $T^y_{init}$, $T^z_{init}$ are part of $T_{init}$.

**Additional details**
- The $n^\circ, m~\mathrm{cm}$ metric, used in Tab. 3, was not mentioned in the "Evaluation metrics" paragraph (L189-194).

**Quality Of The Limitations Section:**

Limitations are addressed clearly

**Questions For Rebuttal:**

- In eq. (1), how are the negatives sampled? Does each term $l_{ij}$ actually include all the negatives in the denominator?
- How long is the training time for each new object?
- Could the authors comment also on the parameter size of the used UCN models? The algorithm requires instantiating an "adaptive" UCN for each new object, which potentially does not scale up well as many new objects are added.
- How exactly is the object diameter $\hat{d}$ estimated?
- In multiple points of the paper, and in particular for the Refinement Module, "geometric features" are mentioned. What exact geometric features were used? In particular, was standard ICP used or somehow additional geometric features/descriptors are used, both for the region proposal module and for ICP?
- L233, Fig. 5b: Could the authors comment on why the increase in accuracy is so sharp and sudden?

**Robotics Focus:**

Highly relevant to robotics but no hardware experiments

**Summary Of Paper:**

The paper proposes an RGB-D approach for few-shot 6-D pose estimation of novel objects, requiring as input a small set of posed RGB-D reference images and also handling severe occlusions. The method effectively combines pre-existing tools, including: 1. the object-agnostic segmentation module UCN, and an InfoNCE contrastive loss, used to extract object masks in both the reference images and in the test one and train an global object feature representation; 2. The Gen6D pose estimator, used to provide an initial 6-D pose estimate from an object crop; 3. ICP for point-cloud based pose refinement. This combination results in improved accuracy over state of the art and increased robustness to occlusions. Additional contributions include:
- The removal of the need for object-centric reference images or object information (e.g., 3-D model or object diameter).
- A method to extract the object crop in a test image which is more robust to occlusion and clutter, by using depth information.

**Summary Of Recommendation:**

While mostly based on a combination of pre-existing tools, the proposed approach advances the performance on the problem of few-shot 6-D pose estimation of novel objects. Furthermore, despite the additional requirement of depth information, it relaxes the assumptions of previous methods and further increases the robustness of these methods against occlusions and clutter, which are practical problems relevant for robotic applications. I therefore believe the paper would be a valuable contribution for the community and recommend acceptance.

---

> ### Author Response · Authors · 2023-08-07
> **Response to Reviewer 4 (L1f1 )**
>
> We are very grateful and thankful for the efforts of the reviewer for making such thorough and constructive comments and suggestions from both the algorithm and the writing skills. We are also encouraged by the reviewer to recognize our work as a valuable and convincing approach with such an in-depth understanding of our paper. Below are our responses to the reviewer's concern:
>
> > Language in writing.
>
> **R4.1** We agree with the reviewer and will rephrase the abstract in the revised version with more straight sentences, and emphasizing the problem w.r.t. open-set and online adaptation while removing the misleading sentences. Also, we will modify the minors mentioned by the reviewer w.r.t. L.124, 128, 203, 225, 246, 250 in the main text and L58, 65 in the supplementary material along with the typos. We want to genuinely thank the reviewer again for the careful proofreading and are really sorry for the typos or any misleading sentences.
>
> > Notation and formatting
>
> **R4.2** We will use $\bigcup_{i=1}^MS_{i}^N$ as recommended by the reviewer and clarify $T_{init}^x$, $T_{init}^y$, and $T_{init}^z$ are part of $T_{init}$.
>
> > Additional details.
>
> **R4.3** An explanation of the metrics used in Tab. 3 will be added. The $n\degree,m\ \mathrm{cm}$ denotes the percentage of accuracy if the prediction error is below $n$ degree and $m$ cm w.r.t. the ground-truth pose.
>
> > In eq. (1), how are the negatives sampled? Does each term actually include all the negatives in the denominator?
>
> **R4.4** For the task on **FewSOL**, yes, all negatives are included in the denominator for each term since the negative samples from each reference are only around 8 (FewSOL includes a single object in each image with a few markers where some markers can be segmented as well). Considering the limited number of the negative samples, it does not add much computing overhead while only 2-5 iterations of adaptation are sufficient to identify the target object. For the task on **LineMOD/LineMOD-OCC/HomeBrewedDB/Wild6D**, since scenes are more complex, particularly for LineMOD-OCC and HomeBrewedDB with multiple and occluded objects, the number of predicted segments from each reference image can be up to 30. **To alleviate the overhead of computation, during each iteration in the denominator, 30~40 negative pairs are randomly sampled from all negative pairs over the reference images**.
>
> > How long is the training time for each new object?
>
> **R4.5** On average, 2-5 online iterations are sufficient for the FewSOL dataset and around 15 iterations for the rest datasets. The required adaptation time to train the adaptive segmentor $\phi^{*}$ is **~15 sec for FewSOL and ~1 min for the rest of datasets** ( which we believe can be further optimized by optimizing the codebase) which also depends on the difficulty of the individual scene (i.e., more difficult scene requires more iterations and vice versa, and the adaptation is finished automatically when the contrastive loss is smaller than the predefined threshold (L.235-237)). After online adaptation, **the inference time for each test image is ~0.75 sec**.
>
> > Could the authors comment also on the parameter size of the used UCN models? The algorithm requires instantiating an "adaptive" UCN for each new object, which potentially does not scale up well as many new objects are added.
>
> **R4.6** This is a good question that was initially not in our concern since we focus more on the online scenario with grasping one specific target object, meaning only one adaptive UCN is instantiated at the same time. We assume the reviewer is asking **when a model is required to identify multiple target objects at the same time**. In this case, multiple adaptive UCNs are essential and a linear increase of VRAM is unfortunately required. However, if the reviewer is asking **when more objects are added to the scenes (e.g., increasing the number of objects from the current setup to hundreds of objects)**, our method is able to scale up without increasing the VRAM by randomly sampling a subset of negative samples while only increasing the number of iterations for the online adaptation. Furthermore, the number of iterations does not need to be linearly scaled up since the adaptive segmentor only focuses on the representation of the target object.
>
> > How exactly is the object diameter estimated?
>
> **R4.7** The canonical (global) object point cloud model is first reconstructed by combining the segments of the target objects from the reference depth images given the ground-truth poses. Afterwards, the object diameter is estimated by measuring the largest diagonal distance of the cube surrounding the object model: $d = \sqrt{|x_{max} - x_{min}|^2 + |y_{max} - y_{min}|^2 + |z_{max} - z_{min}|^2}$, where $x_{max}, y_{max}, z_{max}$ are the max coordinates values of the point cloud model in the x-, y- and z-axis, while  $x_{min}, y_{min}, z_{min}$ are the minimum.

---

> > ### Author Response · Authors · 2023-08-07
> > **Response to Reviewer 4 (L1f1 )**
> >
> > > In multiple points of the paper, and in particular for the Refinement Module, "geometric features" are mentioned. What exact geometric features were used? In particular, was standard ICP used or somehow additional geometric features/descriptors are used, both for the region proposal module and for ICP?
> >
> > **R4.8** We are sorry for the confusion in writing. The geometric features denote the induced global and local point cloud model from reference and test images. The ICP algorithm extracts point-wise FPFH[1] geometric features based on the point cloud. However, there are no additional geometric descriptors involved in our work. We can replace the "geometric features" simply with the "induced point cloud models" if it's better.
> >
> > > L233, Fig. 5b: Could the authors comment on why the increase in accuracy is so sharp and sudden?
> >
> > **R4.9** At the beginning, the adaptive segmentor $\phi^*$ is not able to learn a distinguishable representation of the target object. On the other hand, the online adaptation destroys the initially learned feature space. Therefore, a small performance decrease occurs by randomly predicting the target segment from the scene. As the adaptation goes on, as long as the distinguishable representation of the target object is learned, **the adaptive segmentor $\phi^{*}$ can identify the target object from most test images since all test images are static and under the same data distribution with the reference images ( while a small number of test images are more difficult and require more online iterations which is similar as a long-tail problem)**.  Therefore, a sharp increase occurs at a certain time which can be further improved with more iterations. We also have a thorough explanation of this sharp behavior in L.229-237 in the main text which also includes the procedure of preventing overfitting. We are happy to add more details in the revised version if it's helpful.
> >
> > [1] R. B. Rusu, N. Blodow and M. Beetz, "Fast Point Feature Histograms (FPFH) for 3D registration," 2009 IEEE International Conference on Robotics and Automation, Kobe, Japan, 2009, pp. 3212-3217.

---

> > > ### Comment · Reviewer_L1f1 · 2023-08-14
> > > **Response to the Authors**
> > >
> > > Thank you for your prompt and detailed reply - the answers address my doubts. I would strongly encourage including the details in the manuscript in case of acceptance.
> > >
> > > I have one additional question: Do the test images actually _need_ to be static? That is, is the condition that "all test images are static and [under the same data distribution with the reference images]" a required assumption? What would happen for instance if the robot interacted with the object and as a consequence the latter moved (but the background and the scene stayed the same)? Would the algorithm still be able to estimate the pose of the object after the interaction?

---

> > > > ### Author Response · Authors · 2023-08-14
> > > > **Response to Reviewer 4 (L1f1)**
> > > >
> > > > Dear reviewer,
> > > >
> > > > thank you for your reply. We will add the details in the revision.
> > > >
> > > > Regarding the question, our method does not require the test scene to be fully static, which is also one of the advantages of our method in applications such as continually grasping the target object until it has been successfully grasped. Below are our responses:
> > > >
> > > > - First of all, the rearrangement of the target object does not affect the performance of the base segmentor $\phi$, because $\phi$ does not require any prior knowledge of the object.
> > > >
> > > > - Secondly, since the rearrangement of the target object does not change the properties of the object (texture, color, and shape etc.), as long as the representation of the target object is learned by the updated segmentor $\phi^*$, the identification of the target object from the other objects will not be affected as well, which also holds true for the cluttered scene (e.g., the target object is occluded both before and after the rearrangement in the scene). The reason is that the features of the target object are learned from the partially occluded segmentation of the reference images, where the occluded part of the target object differs among the reference images due to the varying viewpoints. Therefore, the rearrangement of the object can be considered as a different viewpoint of the object with different occluded parts. Thus, the rearrangement of the target object can be considered as predicting a static target object but from a different viewpoint (which has been demonstrated in our work), and does not affect the performance of the updated segmentor $\phi^*$.
> > > >
> > > > - Lastly, the rest of our pipeline (i.e., the region proposal and refinement modules) is not affected by the rearrangement as well.
> > > >
> > > > Since all components of our method are independent of the rearrangement of the target object, our method is able to handle non-static scenes. A future extension of our work could be working on changing and dynamic scenes between the reference and query images, which we believe would be an interesting direction.

---

> > > > > ### Comment · Reviewer_L1f1 · 2023-08-14
> > > > > **Response to the Authors**
> > > > >
> > > > > Thank you for the quick and thorough reply - that answers my question. I confirm my rating.

---

### Official Review · Reviewer_pbhk · 2023-07-19

**Confidence:** 4
**Originality:** Good
**Technical Quality:** Very Good
**Clarity Of Presentation:** Very Good
**Impact:** 2

**Recommendation:**

Weak Accept: I recommend accepting the paper, but will not argue for my recommendation if the majority of other reviewers have a different opinion.

**Review:**

### Strengthes
- The paper is well-written with nice figures and clear equations.

- The proposed method achieves superior object pose estimation performance in various settings while being trained on synthetic data only.

- The proposed method builds upon a simple image segmentation model and basic point cloud registration methods, which is simple yet effective.

### Weaknesses

- The method assumes that the target object is always contained by the test image, what if the object is completely occluded in the test image? Can we tell the absence of the target object from the feature similarity, thus avoiding invalid pose estimation?

- How long does it take to complete the online adaptation? How will the adaptation time scale in the case of large-scene containing hundreds of objects?

**Quality Of The Limitations Section:**

Limitations are addressed clearly

**Questions For Rebuttal:**

Please refer to the weaknesses.

**Robotics Focus:**

Highly relevant to robotics but no hardware experiments

**Summary Of Paper:**

This paper introduces an approach for 6D pose estimation of novel objects with heavy occlusions while not requiring any object information or object-centric images.

More specifically, it fine-tunes the weights of a base image segmentation model using reference images with pose labels, which allows to better localize the target object. Moreover, it leverages the geometric features to improve the robustness against occlusion in comparison to Gen6D.

Experimental results on various benchmarks demonstrated the effectiveness of the proposed method, which could potentially facilitate generalizable 6D object pose estimation in robotic applications.

**Summary Of Recommendation:**

This paper proposes a simple yet effective object pose estimation approach with good representation and convincing results. In particular, it achieves generalizable object pose estimation in heavy occlusions, with models trained on synthetic data only.
This leads to promising robotic applications where the robot needs to interact with novel objects in cluttered scenes.

---

> ### Author Response · Authors · 2023-08-07
> **Response to Reviewer 3 (pbhk)**
>
> First, we would like to genuinely thank the reviewer for the efforts and comments in reviewing our paper and for considering our work as a simple yet effective method. Below are our responses to your concerns:
>
> > The method assumes that the target object is always contained by the test image, what if the object is completely occluded in the test image? Can we tell the absence of the target object from the feature similarity, thus avoiding invalid pose estimation?
>
> **R3.1** This is a very interesting point that is not included in our experiments. In our experiments, we only include the scenarios where the target object is missing or not selected **in the reference images**. In such a case, our model is still robust and able to identify the target object in the test images (details are shown in the supplementary material A.3 and Fig.7). **To the reviewer's concern, identifying the absence of the target object in the test image is possible**:
>
> * In Fig.8, Fig.9, Fig.10 and Fig.11 in the supplementary material, we showed the predictions of the target object with the top-3 confidence scores. In most cases, the confidence score of the target segmentation is over ~0.8 or at least higher than the second candidate with a large margin.  Therefore, **a simple way to identify the absence of the target object is to set a heuristic threshold of the confidence score or the ratio between the highest and the second-highest scores**.
> * However, in some cases, the aforementioned method will fail when an object similar to the target object occurs in the test image. For instance, in Fig.10 (driller), both the highest and second-highest predictions are part of the driller with similar confidence scores, in this case, comparing the ratio of the scores will lead to the absence of the target object which is not true. In contrast, in Fig.9 (duck), although the prediction of duck is higher than the second-highest candidate with a large margin, the confidence score is only ~0.25. Thus, using a high and fixed threshold will lead to a false negative prediction (predicted as an absence of a duck in the test image which is not true).
>
> Overall, **it is possible to identify the absence of the target object in the test image and achieve satisfying results**, but **further fine-tuning of the threshold is necessary to find the optimum**. Combining both the fixed threshold and the ratio between the best two candidates could lead to robust performance, while a higher threshold could lead to higher precision (fewer false positive predictions) but lower recall (more false negative predictions).
>
> > How long does it take to complete the online adaptation? How will the adaptation time scale in the case of large-scene containing hundreds of objects?
>
> **R3.2** The total adaptation time for the online adaptation is **~15 sec** for FewSOL and **~1 min** for the rest datasets ( which can be further optimized by optimizing the codebase), which depends on the complexity of the scene and the number of objects in the scene. After online adaptation, the inference time for each test image is around **0.75 sec**. To alleviate the overhead of computation on large-scale scenes containing hundreds of objects with probably thousands of negative pairs combining the reference images, **a subset of negative pairs (around ~40) are randomly sampled from all negative pairs during each iteration, leading to a scalable solution with similar compute resources (VRAM) as used in our current setup**. More iterations might still be necessary but with less than a linear increase w.r.t. the number of objects in the scene since we only focus on the target object and not all the negative samples are crucial.

---

### Official Review · Reviewer_Zris · 2023-07-19

**Confidence:** 3
**Originality:** Good
**Technical Quality:** Good
**Clarity Of Presentation:** Good
**Impact:** 3

**Recommendation:**

Weak Accept: I recommend accepting the paper, but will not argue for my recommendation if the majority of other reviewers have a different opinion.

**Review:**

Strengths:
+ The paper tackles an interesting setting. Specifically, unlike some prior works which require either object meshes or reference images without clutter, this paper can leverage reference images in a cluttered setting.

+ The empirical results are shown across datasets, and the importance of the contributions is  evaluated well.

Concerns:
- Please correct me if this is wrong, but it seems that reference and test images are from overlapping scenes (i.e. common background and object configurations across reference images and test image). In my view, this is a significant drawback of the setup as it’s not clear if the 6D pose estimation systems or segmentation adaptation actually generalizes (and pays attention to the object vs scene). Moreover, if the reference and test images are indeed the same scene, why do we even need to estimate 6D poses of objects? Instead, we can simply try to obtain relative camera poses between reference and test images (e.g. via COLMAP), and given 6D object pose in any reference image, can trivially obtain 6D pose of object is test image.

- While the paper presents itself as tackling the task of 6D pose estimation, the core technical contributions infact lie in adapting a segmentation model. The 6D pose estimation is actually done via a combination of a previous approach (Gen6D) followed by ICP. The key contribution here lies in improving the detection/segmentation so that clean inputs can be fed to Gen6D and ICP. While this is still a reasonable thing to do, there are no 2D detection/segmentation evaluations. I feel the paper would be better presented (and evaluated) as one for improving few-shot segmentation/detection for a generic object, with one possible application to 6D pose estimation.

**Quality Of The Limitations Section:**

Additional details required

**Questions For Rebuttal:**

a) Please clarify the concern above regarding how the reference images and test images are selected, and what is the variation in the scene (if any) across them.

b) In the ablations, is “SA6D w/o RPM” using the pre-trained segmentor, or is there no segmentor used? If the latter, how is the local region for ICP selected as one should only use points from the forground?

c) On a related note to b), while Gen6D uses only visual cues for prediction, can it not be combined with a refinement model to perform ICP (using an off-the-shelf segmentation)? This would better help disentangle the need for the adaptive segmentation that the paper proposes to learn.

**Robotics Focus:**

Highly relevant to robotics but no hardware experiments

**Summary Of Paper:**

This work tackles the task of 6D pose estimation of a generic object in occluded scene, given a few RGB-D images (under clutter) with known 6D poses for the target object as reference. The approach is to: a) adapt an off-the shelf instance segmentation system to better identify the target object, and b) build a object-specific 3D model using the 6D pose annotations and predicted segmentations. Given a test image, the adapted segmentation system can detect the object and an off-the-shelf 6D pose estimation system can yield initial 6D pose estimate (given object-centric crops from test and reference images), which is then refined via ICP. Experiments show that this approach performs better than prior methods across several examined datasets.

**Summary Of Recommendation:**

First and foremost, I have significant concerns regarding the setup (specifically, whether the train and test images come from common scenes). That said, even assuming these are addressed by the author response, I am still not overly positive about the work and would probably rate this as borderline because the primary contribution seems to be in few-shot adaptation of a segmentation system as opposed to any innovation in 6D pose estimation, which is done via Gen6D + ICP. On average, I am currently indicating my rating as a Weak Reject.
---

Post-discussion update: The author responses (specifically the new experiment showing ability to test in different scenes) did address my concern, and as indicated in my comments in the discussion below, I'd be happy to lean towards acceptance.

Hopefully the authors will include this experiment in the final version.

---

> ### Author Response · Authors · 2023-08-07
> **Response to Reviewer 2 (Zris )**
>
> We truly thank the reviewer for the efforts in reviewing our paper and the valuable comments. Below are our responses to your concerns:
>
> > Please correct me if this is wrong, but it seems that reference and test images are from overlapping scenes (i.e. common background and object configurations across reference images and test images). In my view, this is a significant drawback of the setup as it’s not clear if the 6D pose estimation systems or segmentation adaptation actually generalizes (and pays attention to the object vs scene).
>
> **R2.1** Yes, in our settings, the reference and test images are from the same scene. And to answer the reviewer's concern, we would say the segmentation adaptation actually generalizes to both the target object and the scene (other objects and background etc.). The adaptive segmentor $\phi^{*}$ learns a distinguishable representation of the target object which should in particular work for this scene. However, since the sampled reference images are from different views which include different light conditions (shadows, different direction of light sources), we believe **the learned representation could also generalize to new scenes but with limited variation of the scene.** We genuinely thank the reviewer for the constructive concern and we do believe this is a good direction as a follow-up work where reference and test images are from different scenes with large variations which would require some augmentation or generative process during adaptation to alleviate the domain gap between the reference and test scenes. However, **this work still focuses on the prediction of cluttered novel objects from the same scene, which hasn't been tacked well in the current community**.
>
> > Moreover, if the reference and test images are indeed the same scene, why do we even need to estimate 6D poses of objects? Instead, we can simply try to obtain relative camera poses between reference and test images (e.g. via COLMAP), and given 6D object pose in any reference image...
>
> **R2.2** Theoretically, yes, the 6D pose can be obtained by COLMAP, which is used often to collect datasets. However, there are a few issues with using COLMAP for inference:
> * COLMAP requires more images to reconstruct the scenes depending on the scene complexity. Gen6D uses ~200 images for COLMAP to recover the scene in collecting GenMOP dataset, which is still much simpler than the datasets used in our work. Therefore, we believe more than 200 reference images are needed in our case using COLMAP. In contrast, our method only requires up to 20 reference images. (Sparse reconstruction of COLMAP results also in inaccuracy.)
> * Furthermore, several parameters need to be tuned in COLMAP, e.g., the algorithm used for feature extraction and matching, the threshold of feature matching, the iteration and threshold of RANSAC, the angle of triangulation to build 3D points, the number of iterations for bundle adjustment, image pair and triangulation filtering, post-processing etc. In Gen6D, they also use hand labelling instead of feature matching to avoid mismatching.
> * The processing time of COLMAP is longer and cannot work in real time. In contrast, our method is able to run in real-time after adaptation.
>
> Overall, considering the **complexity, the computation and the sensitivity of parameters**, COLMAP is not suitable in the few-shot scenario for 6D pose estimation, and is not used in prior works either except for the data collection.
>
> > While the paper presents itself as tackling the task of 6D pose estimation, the core technical contributions in fact lie in adapting a segmentation model....
>
> **R2.3** We agree with the reviewer that our main contribution lies in adapting the segmentation. However, the entire research and ideas are developed around generalizing the 6D pose estimation of novel objects under clutter, which also includes the selection of target segments from reference images given the ground-truth pose, the region proposal and refinement modules. Furthermore, **our work does not require any additional training processing using pretrained models, which is particularly important with the emerged foundation models and for the labs without enough computing resources**. While comparing against other few-shot segmentation methods is an interesting topic, it is not the motivation of this work.
>
> > Please clarify the concern above regarding how the reference images and test images are selected...
>
> **R2.4** Instead of selecting reference images from uniformly distributed viewpoints over the scene like Gen6D, in our work, reference images are **randomly** selected from the whole scene, which also increases the difficulty of prediction since reference images do not cover the whole scene, while the rest images are used for testing. The variation would be the changing part w.r.t. the viewpoint, e.g., the light condition, the direction of light sources, the shadows, and the non-occluded part of each object in the scene.

---

> > ### Author Response · Authors · 2023-08-07
> > **Response to Reviewer 2 (Zris )**
> >
> > >  In the ablations, is “SA6D w/o RPM” using the pre-trained segmentor, or is there no segmentor used? If the latter, how is the local region for ICP selected as one should only use points from the foreground?
> >
> > **R2.5** "SA6D w/o RPM" still **uses the pretrained segmentor**, it is an ablation study by only removing the region proposal module (RPM) as indicated in L211-212. Without RPM, the region of interest (ROI) of the target image will be determined by comparing the visual similarity between the cropped reference images and the entire test image (same as used in Gen6D) and results in ambiguous ROI as shown in Fig. 5(c), especially when the target object is occluded.
> >
> > >  On a related note to b), while Gen6D uses only visual cues for prediction, can it not be combined with a refinement model to perform ICP (using an off-the-shelf segmentation)? This would better help disentangle the need for the adaptive segmentation that the paper proposes to learn.
> >
> > **R2.6** Unfortunately, with only an off-the-shelf segmentation is not possible. This is also the motivation of our work to propose adaptive segmentation. The followings are the reasons:
> >
> > * An off-the-shelf segmentation (indicated as $\phi$ in Fig.3) cannot differentiate which object is of interest from clutter **in the test image** but can only predict the segmentations of all objects.
> > * To perform ICP in the refinement module, it is essential to differentiate the segment of the target object in the test image and construct the point cloud from the segmented depth image. Again, since the off-the-shelf segmentation module cannot differentiate the target segment, the proposed adaptive segmentation is vital.
> >
> > We believe there are some differences against the common few-shot segmentation domain, e.g., our work focuses on identifying the target object (segmentation) from the images while the segmentations of all instances are still required, while few-shot segmentation focuses only on segmenting the target instance. Furthermore, although both Gen6D and ICP are existing methods, combining all three modules (online adaptation, region proposal, and refinement modules) together is still novel and demonstrates superior performance, especially without any further training efforts. We kindly hope the reviewer can rethink our contribution.

---

### Official Review · Reviewer_8PhQ · 2023-07-19

**Confidence:** 3
**Originality:** Good
**Technical Quality:** Good
**Clarity Of Presentation:** Very Good
**Impact:** 3

**Recommendation:**

Weak Accept: I recommend accepting the paper, but will not argue for my recommendation if the majority of other reviewers have a different opinion.

**Review:**

- Strength
  - The paper is well-written and easy to follow.
  - The topic of self-adaptative pose estimation is important to many practical applications
- Weakness
  - Need ablation study without online adaptation. Say both pixel label and dense feature are generated with the pretrained UCN.
  - Need baseline results for RePoNet, DualPoseNet on LineMod/LineMOD-OCC, etc.
  - Performance in uncontrolled environment is questionable. As shown in Table 3, on Wild6D, SA6D can hardly out-performs RePoNet, which has a much simpler inference pipeline.

**Quality Of The Limitations Section:**

Additional details required

**Questions For Rebuttal:**

- In Table 1, why SA6D with ICP only performs worse than Gen6D?
- What's the configuration (like the number of pairs and training time) for contrastive learning?
- What's the sensitivity to the performance of the segmentor?

**Robotics Focus:**

Highly relevant to robotics but no hardware experiments

**Summary Of Paper:**

- Online self-adaptation module
  - Extract standard object model: Given reference images and pose of target, a segmentor is employed for segmenting the target objects. Then, by combining the point cloud of the target object, the "standard" object point cloud model is generated.
  - Generate a target-specific segmentor: Given the masks for target/non-target objects, contrastive learning is employed for finetuning the segmentor.
  - Segment test image: Test images are fed into both pretrained and finetuned segmentor for per-pixel label and dense feature, respectively. Target object is selected by comparing the feature with reference.
- Region Proposal Module: Estimate the size of target object based on the standard object model and crop the region of the same size on the target image. Then images are fed into Gen6D to generate a initial guess for the pose.
- Refinement module: Performs ICP between standard object model and segmented object from test image to refine the pose prediction.

**Summary Of Recommendation:**

For now, a weak acceptance is given for the online adaptation. However, more results are required for supporting the effectiveness of the method, especially the effectiveness of the proposed online adaptation module.

---

> ### Author Response · Authors · 2023-08-06
> **Response to Reviewer 1 (8PhQ )**
>
> We genuinely appreciate the reviewer's insightful comments and suggestions in reviewing our paper. Below are our responses to your comments and concerns.
>
> > Need ablation study without online adaptation. Say both pixel label and dense feature are generated with the pretrained UCN.
>
> **R1.1** In our method, we use the pretrained UCN to provide consistent dense features of reference and test images during both training and inference. The idea is to keep the embeddings in the same space. Meanwhile, with the online adaptation, the adaptive segmentor $\phi^{*}$ learns a consistent representation of the target object among different images and reduces the ambiguity of other objects in the scene. Therefore, using only the pretrained UCN ($\phi$) without online adaptation cannot differentiate the target object from others. **To the reviewer's concern**, we conducted the experiment by simply adding iter=0 on LineMOD-OCC/driller, the ADD-0.1d is 0.06, which is similar to iter=1 in Fig.5b, showing the failure of identifying the target object with only the pretrained UCN.
>
> > Need baseline results for RePoNet, DualPoseNet on LineMod/LineMOD-OCC, etc
>
> **R1.2** We understand that the reviewer would like to see more comparisons against RePoNet and DualPoseNet. However, we would like to clarify that our method is category-agnostic and can predict 6D pose on novel objects from unseen categories, which holds true for LatentFusion and Gen6D as well. However, **RePoNet and DualPoseNet work only for category-level 6D pose estimation and cannot predict on LineMod/LineMOD-OCC/FewSOL/HomeBrewedDB using novel categories.** Therefore, the comparisons against RePoNet and DualPoseNet are conducted on a category-level dataset (Wild6D), where RePoNet and DualPoseNet are trained within the same categories although our method is not trained on this dataset.
>
> > Performance in uncontrolled environment is questionable. As shown in Table 3, on Wild6D, SA6D can hardly out-performs RePoNet, which has a much simpler inference pipeline.
>
> **R1.3** Our method focuses on a more generic case where we remove the constraints of category-level estimation. We would like to note to the reviewer that, **our method is not trained on any objects from Wild6D dataset while RePoNet is trained and tested on the same dataset (Wild6D)**. Therefore, this is an unfair comparison of our method. However, we showed that our method can still achieve competitive performance and even better results using the stricter metrics ($5\textdegree2cm$ and $5\textdegree5cm$).
>
> > In Table 1, why SA6D with ICP only performs worse than Gen6D?
>
> **R1.4** SA6D with ICP performs worse than Gen6D on non-occluded dataset (LineMOD) or less occluded objects (cow and rabbit in HomeBrewedDB) while still better on more occluded objects (LineMOD-OCC and flange/car in HomeBrewedDB). The reason is without the Region Proposal Module (RPM) and the following Refinement Module (RFM), using ICP solely always leads to a local solution (or local optima considering the pose estimation as an optimization problem). ICP only uses the constructed point cloud from depth image while Gen6D uses the visual information from RGB, which also explains why Gen6D performs better when the target object is less occluded because it is easier for Gen6D to identify and localize the target object.
>
> > What's the configuration (like the number of pairs and training time) for contrastive learning?
>
> **R1.5** For contrastive learning (the so-called online adaptation procedure), everything is conducted during inference, i.e., **no training time is needed**. We assume the pairs indicated by the reviewer denote both the positive and negative pairs used during contrastive learning. **For the positive pairs**, the number depends on the the number of reference images, which is 8 for FewSOL and 20 for the rest datasets. Meanwhile, **the number of negative pairs** depends on the number of segments predicted by the pretrained UCN from each reference image (e.g., for simple scenes in FewSOL could be around 8 negative pairs from each reference image and around 15-30 in LineMOD-OCC from each reference image). **To alleviate the overhead of computation**, during each iteration, 30~40 negative pairs are randomly sampled from all negative pairs over the reference images. **The total adaptation time** for the online adaptation is ~15 seconds for FewSOL and ~1 min for the rest datasets ( which can be further optimized by optimizing the codebase). After online adaptation, the inference time for each test image is around 0.75 sec.
>
> > What's the sensitivity to the performance of the segmentor?
>
> **R1.6** Our method doesn't require accurate segmentation although it is still crucial, since the object model can be reconstructed and optimized by the point density over the reference images. Furthermore, our method is robust against negative positive samples, please refer to our **supplementary material A.3 and Fig. 7** for details.

---

> > ### Comment · Reviewer_8PhQ · 2023-08-07
> > **Response to Authors**
> >
> > Thank you for addressing my concern. Indeed, the proposed method SA6D is designed for few-shot adaptation, thus requiring SA6D to achieve similar accuracies to other non-few-shot methods is not fair.
> >
> > Based on your reply, in my understanding, for every new object assigned for pose estimation, SA6D requires around 1 minute for initialization, and after initialization, SA6D takes 0.75 sec/image for pose estimation. Is this correct? What device is used for collecting these profiling results?

---

> > > ### Author Response · Authors · 2023-08-08
> > > **Response to Reviewer 1 (8PhQ )**
> > >
> > > Dear reviewer,
> > >
> > > Thank you for your reply and thank you for understanding that it is not a fair comparison of our work with RePoNet/DualPoseNet on LineMOD/LineMOD-OCC, etc.
> > >
> > > In case the "initialization" mentioned by the reviewer is indicating the "online adaptation" process, yes, it takes ~15 sec (2-5 online iterations) for FewSOL dataset and ~1 min (15 online iterations) for the other used datasets. The online adaptation is using only the reference images and the time of online adaptation depends on the number of required online iterations. Thus, more complex scenes require more online iterations. After online adaptation, the inference time on each test image takes ~0.75 sec. We would like to also clarify that we haven't really focused on optimizing the running time in our work. Therefore, we believe the running time can be further reduced after cleaning the code and removing redundant tensors generated during online adaptation.
> > >
> > > We use **a single Nvidia V100 GPU** to collect these results.

---

### Comment · Reviewer_Zris · 2023-08-11
**Unconvinced about the setup**

I would like to thank the authors for their responses. However, the responses actually affirmed my concerns about the setup and I would be inclined to argue strongly against acceptance. Given the other reviewers are more positive, I wanted to raise this as a common discussion point and would like to hear from the other reviewers/authors if they can help address these fundamental concerns about the setup.

Specifically, the paper tackles a setting where K (say 20) images depicting an object in a specific scene (with known 6D object pose of the object in each image) are available.  Then, given a new image in the same scene (i.e. same background, unchanged configuration of surrounding objects), the proposed approach seeks to predict the 6D pose of the object in this new image. First, I am not sure why this is a valid/useful setup to consider. Secondly, this task can also be equivalently be framed as that of camera registration and I think the paper needs to compare to relevant baselines. I appreciate the authors point about possible generalization to new scenes, but without empirical evidence that this is feasible, I am inclined to only consider the 'within-scene' setup as one that the approach can be applied to and this raises some key issues:

1) I fail to see practical settings where this setup would be useful. The assumed input is several images of an object in a specific scene with known 6D poses of the object, and the query image is also from the same scene. This is in contrast to other setups which could be more useful e.g. a) given 3D mesh of object, estimate 6D pose in novel scenes or b) given reference images with known poses across a few scenes, estimate 6D pose in a novel scene. Given this approach requires reference and query images in the same scene, this implies that the user is be able to obtain 6D poses of the object for the K images in the scene of interest but not for the query (K+1st) image and I am not sure what scenario would arise with such constraints.

2) Given that the scene is unchanged, the knowledge of 6D poses of the object in the reference images implies known relative cameras for all the reference images. Thus, given a new image, inferring it’s camera pose can equivalently yield the 6D pose for the target object. In my view, this necessitates comparisons camera registration/SfM methods. While the authors are correct that COLMAP requires many more images for a dense reconstruction, I believe the recent advances in improved features/learned matching make it more robust (specifically if we only require cameras poses and not a reconstruction). Moreover, the task here is only registration/localization of a new camera given other calibrated cameras (as opposed to SfM where no camera extrinsics are known). Given the equivalence of the two tasks, I do not think this work can be accepted without such comparisons (e.g. using the localization code in https://github.com/cvg/pixel-perfect-sfm, https://github.com/cvg/Hierarchical-Localization).

---

> ### Author Response · Authors · 2023-08-13
> **Response to the reviewer 2 (Zris)**
>
> Dear reviewer,
>
> we are grateful for your constructive discussion and comments. Below are our responses:
>
> After carefully checking the reviewer's concerns, we found there is a crucial **misunderstanding** between the reviewer and us when we use the term of "same scene". We are sorry if there's any confusion caused by our answer. To clarify this, the "same scene" now refers to using exactly the same scene without changing the position of objects or any configuration of the environment. **Our method does not require any assumption of using the reference and query images from the same scene**. In contrast, **our method can be applied when the reference and query images are selected from different scenes with rearrangement of objects**. On the other side, the reason that prior work [1,2,3] and our work seem to use the same scene for both reference and query images is that **the existing public datasets are recorded with limited variation of the scene (e.g., the rearrangement of the objects)**. Thus, **we need to emphasize that using the same scene is essentially the limitation of the public datasets rather than the limitation of prior and our methods**.
>
> Nevertheless, to address the reviewer's concerns, **we have conducted the experiments where we sample 20 images from LineMOD-OCC (LMO) as reference, and evaluate query images from LineMOD (LM) which includes different background and objects**. Thus, the reference and query images are from different scenes with changing light conditions and configuration of surrounding objects. We still achieve competitive results compared to the original setup (first line in the table). Note that this is still not a fair comparison for our method since we use reference images with occluded target objects, which makes it more difficult to reconstruct the object model. However, this experiment demonstrates the capability of our method on query images from novel scenes. **Examples are uploaded and will be added in the revision**.
>
> | Ref. image | Query image | eggbox | duck | cat | glue | avg. |
> |:-:|:-:|:-:|:-:|:-:|:-:|:-:|
> | LM |  LM | 0.73 | 0.73 | 0.47 | 0.72 | 0.66 |
> | LMO | LM | 0.70 | 0.71 | 0.46 | 0.72 | 0.64 |
>
> Based on the clarification, we would like to answer the reviewer's concerns in detail:
>
> - We would like to inform the reviewer that the few-shot 6D pose estimation of a novel object (especially for category-agnostic objects) is still an open and under-explored topic that was proposed just in recent years. **Our work follows the same setup proposed by prior work such as LatentFusion [1], Gen6D [2], and FS6D[3] and use the same public datasets, which are essential to achieve fair comparisons.** Moreover, our method essentially improves practical use by removing the constraints of prior work.
>
> - The reviewer mentioned that estimating 6D pose in novel scenes given the 3D mesh of objects could be more useful, **which is exactly what our work can achieve**. Our method essentially reconstructs the 3D model of the object from reference images and is capable of evaluating the query images from new scenes with rearrangement of objects. Furthermore, our method improves the generic use case without requiring the 3D mesh of a novel object. In addition, **our method is also able to work with reference images across a few scenes**. The only additional requirement is that the poses of reference images from different scenes need to be aligned in the canonical coordinates in order to reconstruct the canonical object model.
>
> - The reviewer stated that he is not sure what scenario would arise such constraints about obtaining the 6D poses of the object in the query images if obtaining the poses of reference images is possible. We believe this question is caused by the misunderstanding of the "same scene" and the misunderstanding of the limitation of our methods. We hope this question is addressed after the clarification. In particular, **our method only requires labelling of reference images once, and can predict on new scenes without further labelling**.
>
> - Regarding the practical uses, the setup can be used for AR/VR, i.e., replacing the target object in new scenes with another synthetic generated object given the predicted 6D pose and the detected segmentation of the target object. The setup can also be used for item picking in the industrial line, where the light condition and picking environment do not change significantly, however, the objects might change often depending on the different products. For such cases, human labelling is only required once to obtain the posed reference images while a model can adapt quickly to new objects in the product lines.
>
> [1] Keunhong et al. “LatentFusion: End-to-End Differentiable Reconstruction and Rendering for Unseen Object Pose Estimation.” CVPR, 2020.
>
> [2] Liu et al. “Gen6D: Generalizable Model-Free 6-DoF Object Pose Estimation from RGB Images.” ECCV, 2022.
>
> [3] He et al. “FS6D: Few-Shot 6D Pose Estimation of Novel Objects.” CVPR, 2022.

---

> > ### Author Response · Authors · 2023-08-13
> > **Further response to the reviewer 2 (Zris)**
> >
> > Regarding the second concern of the reviewer:
> >
> > > 2. Given the equivalence of the two tasks, I do not think this work can be accepted without such comparisons (e.g. using the localization code in https://github.com/cvg/pixel-perfect-sfm, https://github.com/cvg/Hierarchical-Localization).
> >
> > The reviewer stated that the prediction of the 6D pose of the target object given the pose reference images is equivalent to the prediction of the camera pose of the query image given the calibrated cameras (camera registration/localization) using SfM methods. In particular, the reviewer claimed that our work should not be accepted without comparing against SfM methods.
> >
> > We believe that **this concern is still caused by the misunderstanding**, and therefore we politely disagree with the reviewer's opinion for the following reasons:
> >
> > - Firstly, as mentioned in the above response, our method is capable to predict on query images from novel scenes, while **the camera registration mentioned by the reviewer is only working on the same scene**. It is also the reason that none of the mentioned related work ( in section 2 of our paper including both category-agnostic and category-level few-shot 6D object pose estimation) compares against SfM methods. The only work related to SfM is also mentioned in L. 87-90 where OnePose and OnePose++ build the object model from a single RGB video sequence using SfM. However, they require hundreds of images and thus are out of the scope of few-shot learning. Furthermore, to reconstruct the model of the target object from clutter using SfM, a target-aware object detector is still required to predict the region of interest of the object. In addition, it is essential to reconstruct the object model rather than only the camera registration in order to generalize on novel scenes.
> >
> > - Apart from the purpose of working on query images from novel scenes, **our task is still not equivalent to camera registration/localization**. In our task, the objective is to estimate the location and the 6D pose of the target object from clutter, which can be used for robotic grasping or AR/VR. For instance, robotic grasping requires a 3D bounding box of the object (see Fig. 4 in the main paper and Fig. 13-17 in the supplementary material), **which requires estimating the model or the size of the object**. For AR/VR, to replace the target object with another synthetic object, **the segmentation of the target object is also required to be predicted** (see Fig.8-11 in the supplementary material). Our method can satisfy all the requirements while the camera registration only predicts the viewpoint of the camera w.r.t. the entire scene, **but without any information w.r.t. the target object (e.g., where is the object? How large is the object? How much is the object occluded?). Without predicting the information w.r.t. the target object, a target-oriented robotic grasping or AR/VR application is not possible.**
> >
> > In summary, we think the reviewer's concerns are mainly caused by the misunderstanding between us in terms of the "new scene". We also demonstrated that our method is capable of predicting the query images from novel scenes, e.g., with the rearrangement of objects, changing background and configuration. Based on the constructive discussions from the reviewer, a future extension can focus on how to learn a consistent and stable object representation from reference images across multiple and more challenging scenes.

---

> > > ### Comment · Reviewer_Zris · 2023-08-15
> > > **Thank you for the additional experiments**
> > >
> > > Thank you for providing the additional experiments. Given that these do show that the method can indeed handle scene variation, I would not argue against the paper as I find this setup more convincing.
> > >
> > > Would it be possible to please share some sample reference/query images of the object in the LMO/LM setup above to understand if there is indeed some meaningful variation? Assuming there is, I would request the authors to please include this in the final version.

---

> > > > ### Author Response · Authors · 2023-08-15
> > > > **Examples uploaded**
> > > >
> > > > Dear reviewer,
> > > >
> > > > Thank you for your quick answer. We have uploaded the examples in a PDF file. Please find it below your review since OpenReview does not support uploading files under comments. Results will also be added in the final version.

---

> > > > > ### Comment · Reviewer_Zris · 2023-08-15
> > > > > **Concern Addressed**
> > > > >
> > > > > Thank you for providing these. It is reassuring to see the approach work despite rearrangements. I'd be happy to update my rating to lean towards acceptance.

---

### Decision · Program_Chairs · 2023-08-30

**Decision:**

Accept (Poster)

**Comment:**

The paper introduces a novel method on few-shot object pose estimation that can handle novel objects.

After the rebuttal, most concerns of the reviewers have been addressed, and all the reviewers agree on accepting the paper.

I recommend the paper as a poster presentation due to the lack of applying the pose estimation method to real robotic applications such as grasping or manipulation of objects.